



# Complementary aerosol mass spectrometry elucidates sources of wintertime sub-micron particle pollution in Fairbanks, Alaska, during ALPACA 2022

Amna Ijaz[1,2], Brice Temime-Roussel[1], Benjamin Chazeau[1], Sarah Albertin[3], Stephen R. Arnold[4], Brice Barret[5], Slimane Bekki[6], Natalie Brett[6], Meeta Cesler-Maloney[7], Elsa Dieudonne[8], Kayane K. Dingilian[9], Javier Fochesatto[10], Jingqiu Mao[7], Allison Moon[11], Joel Savarino[3], William Simpson[7], Rodney J. Weber[9], Kathy S. Law[6], Barbara D'Anna[1]

[1]Aix-Marseille Université, CNRS, LCE, Marseille, 13005, France
[2]Present Address: Atmospheric, Climate, & Earth Sciences Division, Pacific Northwest National Laboratory, Richland, Washington 99354, United States
[3]University of Grenoble Alpes, CNRS, IRD, Grenoble INP, INRAE, IGE, F-38000 Grenoble, France
[4]School of Earth and Environment, University of Leeds, Leeds, LS2 9JT, United Kingdom
[5]Laboratoire d'Aérologie, Université Toulouse III-Paul Sabatier, CNRS, Toulouse, France
[6]Sorbonne Université, UVSQ, CNRS, LATMOS-IPSL, Paris, France
[7]Department of Chemistry and Biochemistry and Geophysical Institute, University of Alaska, Fairbanks, AK, United States
[8]Laboratory of Physics and Chemistry of the Atmosphere, University of Littoral and Opal Coast, Dunkerque, France
[9]School of Earth and Atmospheric Sciences, Georgia Institute of Technology, Atlanta, Georgia 30332, United States
[10]Department of Atmospheric Sciences, University of Alaska, Fairbanks, AK, United States
[11]Department of Atmospheric Sciences, University of Washington, Seattle, Washington 98195, United States

*Correspondence to*: Barbara D'Anna (barbara.danna@univ-amu.fr) and Amna Ijaz (amna.ijaz@pnnl.gov)

Fairbanks, Alaska, is a subarctic city that frequently suffers from non-attainment of national air quality standards in the wintertime due to the coincidence of weak atmospheric dispersion and increased local emissions. However, significant uncertainties exist about aerosol sources, formation, and chemical processes during cold winter periods. We aim to determine the composition, size, and concentrations of atmospheric sub-micron non-refractory particulate matter (NR-PM$_1$) and quantify their sources in the urban centre of Fairbanks. As part of the





Alaskan Layered Pollution and Chemical Analysis (ALPACA) campaign, we deployed a Chemical Analysis of Aerosol Online (CHARON) inlet coupled with a proton transfer reaction - time of flight mass spectrometer (PTR-ToF MS) and an Aerodyne high-resolution aerosol mass spectrometer (AMS) to measure organic aerosol (OA) and NR-PM$_1$, respectively. We used positive matrix factorisation (PMF) for source identification. PTR$_{CHARON}$ factorisation delineated four residential heating sources, including wood and oil combustion, that contribute 47 ± 20% and 16 ± 9% of OA$_{CHARON}$, on average, respectively. In contrast, only a single biomass burning-related factor was identified by AMS for both OA and NR-PM$_1$, but it provided information on two additional factors that were rich in sulphur and nitrate. These results demonstrate that PTR$_{CHARON}$ can generate robust quantitative information with enhanced resolution of organic aerosol sources. When combined with suitable complementary instruments like the AMS, such evidence-based insights into the sources of sub-micron aerosol pollution can assist environmental regulators and citizen efforts for the improvement in air quality in Fairbanks and in the wider Arctic winter.

**Keywords** PM$_1$, mass spectrometry, source apportionment, Fairbanks, Arctic, air quality, CHARON PTR-ToF MS, HR-ToF AMS, proton transfer reaction

# 1 Introduction

Extremely cold urban regions of the Earth, such as in the Arctic, experience poor dispersion of atmospheric pollution, especially during the wintertime, when the unique meteorological characteristics, such as extremely low solar radiation and strong radiative cooling at the surface, are coupled with enhanced local anthropogenic emissions from heating, industry, and transport. A good example is the subarctic city of Fairbanks, Alaska, where air quality standards are frequently violated during the winters with concentrations of fine particulate matter (i.e., with aerodynamic diameters smaller than 2.5 µm; PM$_{2.5}$) exceeding the 24-h limit of 35 µg/m$^3$ defined by EPA's National Ambient Air Quality Standards (Dunleavy and Brune, 2020; EPA, n.d.). Not only is Fairbanks one of the cities with the most polluted wintertime air in the US, but it has also been declared a 'moderate non-attainment area' since 2009 and due to the persistence of the problem, it was reclassified as a 'severe non-attainment area' in 2017. Increased local anthropogenic emissions and poor atmospheric dispersion due to strong surface-based temperature inversions (> 0.5°C/m in the lowest 10 m above the ground) are major causes of wintertime pollution in the region (Tran and Mölders, 2011; Mayfield and Fochesatto, 2013). Many research studies have recognised biomass combustion as the major source of aerosol in Fairbanks (Ward et al., 2012; Wang and Hopke, 2014; Kotchenruther, 2016; Ye and Wang, 2020; Haque et al., 2021) that drives overall PM$_{2.5}$ concentrations across the city during strong temperature inversion conditions (Robinson et al., 2023). A comprehensive study covering three winters from 2008–2011 apportioned 60–80% of PM$_{2.5}$ mass at four locations in Fairbanks to emissions from residential wood stoves, open burning of biomass, outdoor boilers, and other solid-fuel combustion. (Ward et al., 2012). Source apportionment of year-round PM$_{2.5}$ in the past two decades [2008–2009 (Haque et al., 2021), 2005–2012 (Wang and Hopke, 2014), 2009–2014 (Kotchenruther, 2016), and 2013–2019 (Ye and Wang, 2020)] also revealed woodsmoke as a major contributor to PM$_{2.5}$ loads [47.5% (Haque et al., 2021), 40.5% (Wang and Hopke, 2014), ~52% (Kotchenruther, 2016), and ~19% (Ye and Wang, 2020)]. Wildfire activity and residential wood





combustion are the major sources in summer and winter, respectively. The persistent role of wood-burning emissions in shaping the air quality of Fairbanks during winters triggered the implementation of a two-stage burn restriction in 2015 by the Alaska Department of Environmental Conservation (ADEC). The ADEC advisories restricted the operation of solid-fuel heating devices and required alternative heat sources to be used on days with weak atmospheric dispersion and $PM_{2.5} > 25$ µg/m$^3$ are observed or forecasted (Fye et al., 2009; Czarnecki, 2017; Jentgen, 2022). Sulphate has been observed to be the second largest component of $PM_{2.5}$ mass in Fairbanks (Ward et al., 2012; Wang and Hopke, 2014 ), forming ~33% of the annual average $PM_{2.5}$ mass (Ye and Wang, 2020). Isotope analyses have revealed 62% of this $PM_{2.5}$ sulphate to be primary (e.g., from residential heating oil combustion) during the winters (Moon et al., 2023).

The aforementioned studies on air quality in Fairbanks have focused on $PM_{2.5}$; however, in many large cities of the world, $PM_1$ (i.e., aerodynamic diameters smaller than 1 µm; $PM_1$) constitutes 75–80% of $PM_{2.5}$, and it is recognised as the major cause of negative health effects of atmospheric aerosol (Wang et al., 2015; Mainka and Zajusz-Zubek, 2019) due to its capability to spread deeper into the respiratory or cardiovascular systems (Meng et al., 2013; Liu et al., 2013; Chen et al., 2017). Currently, $PM_1$ concentrations are not regulated globally, but its strong contribution to atmospheric $PM_{2.5}$ loads and impacts has implications for the attainment of the latter's regulatory limits. Efforts to monitor $PM_1$ are surprisingly scarce, even in a 'non-attaining' city, such as Fairbanks. Characterising the chemical composition of such sub-micron atmospheric aerosol and capturing the variation in their mass concentrations is key to unravelling the complexities of local emissions and their transformation in Fairbanks and, most importantly, to underscore the health and policy implications of atmospheric emissions.

Mass spectrometric techniques have advanced over the years, now featuring greater mass accuracies, resolving powers, and sensitivities. For instance, the Aerodyne high-resolution time-of-flight aerosol mass spectrometer (HR-ToF AMS; called AMS from hereon) is a well-established method for quantification of non-refractory NR-$PM_1$. Aerosol vapourisation at high temperatures and electron ionisation result in substantive molecular decomposition, facilitating quantification with high time resolution (Decarlo et al., 2006), but at the cost of molecular-level resolution. The lack of molecular-level information provided by AMS encourages the use of complementary techniques to better understand both primary aerosol sources and secondary aerosol formation. For instance, extractive electrospray ionisation (EESI)-ToF MS has been successfully deployed in Beijing (Tong et al., 2021) and in Zurich to resolve multiple OA sources (Stefenelli et al., 2019a; Qi et al., 2019). Although the EESI-ToF MS provides molecular-level information in detail, its quantitative response is variable and selective for polar species, preventing its independent application for ambient measurements. Other measurement methods, such as thermal desorption aerosol GC/MS flame ionisation detector (TAG)(Williams et al., 2006) and filter inlet for gases and aerosols chemical ionisation (FIGAERO-CIMS)-ToF MS (Lopez-Hilfiker et al., 2014) similarly offer better chemical resolution than the AMS, but a lower temporal resolution. Semi-continuous measurements, such as those from TAG and FIGAERO-CIMS, may not capture the rapid variation in sources.

To improve the analysis of sub-micron OA in ambient air, a novel inlet system called the chemical analysis for aerosol online (CHARON) was developed to collect real-time measurements (Eichler et al., 2015). This CHARON





inlet minimizes thermal and ionisation-induced fragmentation of sampled OA by employing a low-temperature vapourisation system (150°C $\leq$) and coupling with a relatively softer and less selective ionisation method, such as the proton-transfer reaction (PTR). The CHARON PTR-ToF MS (called PTR$_{CHARON}$ from hereon) was first successfully used for the characterisation of OA from ship exhaust (Eichler et al., 2017), followed by quantification of ambient OA in Lyon, France, and Valencia, Spain, and OA source apportionment in Innsbruck, Austria (Müller

et al., 2017). Recently, it was used to quantify individual compounds in laboratory-generated secondary organic aerosol (Lannuque et al., 2023) and complex mixtures, such as vehicular gasoline emissions and atmospheric organic matter (Piel et al., 2019; Kostenidou et al., 2024). Additionally, the analyser (commonly a PTR-ToF MS) coupled to the CHARON inlet can measure gas-phase species as well, creating the opportunity to explore VOC precursor emissions or phase partitioning (Peng et al., 2023; Gkatzelis et al., 2018). Overall, PTR$_{CHARON}$ and AMS

are complementary techniques that provide robust qualitative and quantitative information. The former features good molecular resolution of the OA in contrast to the AMS, but has limited ability to analyse particles smaller than 150 nm (Eichler et al., 2015); the latter instrument covers smaller aerosol (i.e., > 60 nm) and detects inorganic components too (Decarlo et al., 2006). Therefore, together, they provide an excellent combination of real-time and quantitative data on atmospheric ambient aerosol.


    The role of sub-micron aerosol pollution in the deterioration of air quality in Fairbanks – and other anthropogenically influenced regions of the wider Arctic – is not understood well. To address this issue we deployed a PTR$_{CHARON}$ and an AMS in the urban centre of Fairbanks during the ALPACA (Alaskan Layered Pollution and Chemical Analysis) (Simpson et al., 2024) campaign as part of the French CASPA (Climate-Relevant

Aerosol Sources and Processes in the Arctic) project in January–February 2022. We aimed to determine the composition, concentrations, and sources of atmospheric NR-PM$_1$. In this paper, we present: (i) an intercomparison of the performance of the two instruments focusing on OA quantitation, (ii) the identification of major OA sources in Fairbanks and their variation during the field campaign, and (iii) the source apportionment of organic and inorganic aerosol (e.g., ammonium, nitrate, and sulphur). The results obtained here demonstrate that a good mass

resolution, combined with the soft ionisation of the PTR$_{CHARON}$, provides both qualitative and quantitative data and allows a better understanding of NR-PM$_1$ sources. This knowledge is a key motivator for policy and citizen efforts to prevent and control air pollution, not only in Fairbanks, but also across other regions in the Arctic given the similarities in weather and climate regime.

## 2 Methodology

**2.1 Field campaign**

    The data presented in this study were collected during the ALPACA campaign in Fairbanks, Alaska, US from January 20 to February 26, 2022. ALPACA is an international collaborative field experiment to understand sources of outdoor and indoor air pollution in the cold and dark conditions of Fairbanks' winter. The scientific objectives and broad preliminary findings of the experiment were recently reviewed (Simpson et al., 2024). All instruments

used for this study were housed in a trailer parked at the Community and Technical College (CTC) of the University





of Alaska, Fairbanks (64.84064°N, 147.72677°W; 136 m above sea level). The CTC is in the urban core of Fairbanks, close (within 40 m) to a major downtown road and parking area (Simpson et al., 2024); the west of this locality is dominated by residential activities, while the north and east have commercial activity.

The trailer was equipped with a suite of particle counters and mass spectrometers that record data at high temporal resolutions (varying from 10 seconds to 2 minutes). A scanning mobility particle sizer (SMPS) and a multi-angle absorption photometer (MAAP) were utilised to measure the distribution of particles sized 15.1 to 661.2 nm and black carbon concentrations, respectively. Two mass spectrometers, PTR$_{CHARON}$ (150~1000 nm) and AMS (60~1000 nm), were connected to the same inlet that sampled air at 3.5 meters above ground level through a short
($\approx$ 1 m) stainless tube with a 1/2" outer diameter extending through the trailer roof. A HEPA filter was placed upstream of the inlet at regular intervals (twice a week) to measure the instrumental background. Additionally, meteorological data, including ambient temperatures at 3 and 23 m; wind speed and direction; and trace gases, namely CO, $SO_2$, $O_3$, NO and $NO_2$, were recorded as described in a previous study associated with the campaign (Cesler-Maloney et al., 2022).

**2.2 Instrumentation**

**2.2.1 PTR-ToF MS: Operation and data processing**

OA was quantified with a PTR-ToF MS (PTR-TOF 6000 X2, Ionicon Analytik GmbH, Austria) coupled to a CHARON inlet in near real-time at 20-sec temporal resolution, i.e., the PTR$_{CHARON}$. The CHARON inlet has been described in detail by Eichler et al (Eichler et al., 2015) and its applications were further evaluated and improved
in subsequent studies (Müller et al., 2017; Leglise et al., 2019; Müller et al., 2019; Piel et al., 2019; Peng et al., 2023). Here, the PTR-ToF MS was configured to alternate between direct sampling of air to measure VOCs for 15 minutes (not included in the current study) and sampling of particulate matter through the CHARON inlet for 45 minutes. The instrument was operated at a low E/N of 65 Td (i.e., drift voltage/pressure; pressure, temperature, and voltage of the drift tube were set at 2.6 mbar, 120°C, and 265 V) and in RF mode for optimal sensitivity. Raw data
was obtained as described in **Section S1** and pre-processed with the Ionicon Data Analyzer (IDA, version 1.0.0.2), followed by post-processing (i.e., background subtraction, conversion of raw signal to mixing ratios, temporal averaging, PMF input generation) with an in-house data processing tool, PeTeR Toolkit (version 6.0; Igor 6.37). The error matrix was also calculated by PeTeR using uncertainties in ion counts and background signals. Among the 1118 ions resolved, 336 were retained above the S/N, and 318 ions could be given a molecular formula based
on the criteria described in **Section S2**. PTR ToF MS records raw signals in counts per second (cps) that were converted to mixing ratios according to the molecular identity determined for the detected ions and their protonation efficiencies (further details in **Section S1**). For comparison with the AMS, mixing ratios were converted to mass concentrations, i.e., µg/m$^3$, using **Equation S2**. Mass concentrations calculated for the PTR$_{CHARON}$ require a critical correction for the enrichment of sampled OA in the aerodynamic lens of the CHARON inlet (Eichler et al., 2015;
Müller et al., 2017); further details are provided in **Section S3**. Total (or bulk) OA at a given point in time was the sum of mass concentrations of all ions, which was corrected for fragmentation using a previously reported method (Leglise et al., 2019), which increased the total OA mass concentrations by 17%.



Species with $m/z > 50$ (the largest $m/z$ detected above the S/N was 425) were retained for PMF of OA. Smaller
molecules of $m/z$ 18–50 were present in low concentrations; they are expected to be too volatile to be present in
OA and were likely detected by PTR$_{CHARON}$ as artefacts from the denuder function. Time series were averaged to 2
minutes (from 20 seconds) and two matrices ($m/z \times$ time points) were extracted: (i) ion concentrations and (ii) their
measurement uncertainties, using PeTeR version 6.0 in Igor 6.37. The final matrices – after removing empty rows
and columns – had the following dimensions $336 \times 17{,}986$. Where required, ion intensities (in either ppb or µg/m$^3$)
were normalised to the sum of all measured intensities.

### 2.2.2 Operation and data processing of the AMS

NR-PM$_1$ were monitored by an AMS (Aerodyne Research Inc., Billerica, USA) with spectral acquisition at 1-
minute intervals. The instrument has been described previously (Decarlo et al., 2006; Canagaratna et al., 2007).
Briefly, ambient particles are sampled through a critical orifice, focused into a narrowed beam by an aerodynamic
lens, accelerated toward a heated element (600°C) for flash vapourisation, and then ionised by electron impact (70
eV at 10$^{-7}$ torr). Finally, the ions are analysed by a time-of-flight mass spectrometer. Standard calibrations were
performed using 300 nm size-selected dried ammonium nitrate and ammonium sulphate particles at the beginning
and the end of the campaign. Nitrate-equivalent values of sample mass concentrations were converted by applying
relative ionisation efficiencies (RIEs) for organics, nitrates, ammonium, sulphur, and chlorides (1.4, 1.1, 3.15, 1.93,
and 1.3, respectively). For quantitative purposes, the collection efficiency (CE) of particles must be considered as
strongly viscous particles in the sampled air are prone to bouncing off the vapouriser, thereby suffering from
reduced detection. We used the time series of composition-dependent CE (CDCE) generated by PIKA using a
previously reported algorithm (Middlebrook et al., 2012), which ranged from 1.00 to 0.35.

Data was averaged to 2 minutes and extracted as concentration and measurement uncertainty matrices ($m/z \times$ time
points) using SQUIRREL version 1.65 and PIKA version 1.25 in Igor 8.04. Separate matrices (and subsequently
PMF) were prepared for organic only (abbreviated AMS$_{org}$) and by combining organic and inorganic species
(abbreviated AMS$_{org+inorg}$). The inorganic species included in the analyses were nitrates ($m/z$ 30, NO$^+$ and 46, NO$_2^+$),
sulphur ($m/z$ 48, SO$^+$; 64, SO$_2^+$; 80, SO$_3^+$; 81, HSO$_3^+$; and 98, H$_2$SO$_4^+$), ammonium ($m/z$ 15, NH$^+$; 16, NH$_2^+$; and
17, NH$_3^+$), and chlorides ($m/z$ 35, Cl$^+$ and 36, HCl$^+$). Error matrices were calculated by PIKA based on uncertainty
in ion counts, background signal, air beam correction, and electronic noise (Sueper, 2014). Atomic O/C and H/C
ratios were calculated based on established methods (Aiken et al., 2007; Aiken et al., 2008; Canagaratna et al.,
2015). Where needed for comparison with the PTR$_{CHARON}$, mass concentrations of PAHs were estimated from
fragments as described previously (Herring et al., 2015), and levoglucosan was estimated as detailed in **Section S4**.

Species with $m/z$ 12–120 were retained for PMF in this study, excluding important PAHs detected up to $m/z$ 252;
such PAHs were used as external tracers for factor identification. All PAHs were included in total OA quantification
and associated comparisons. This exclusion is expected to cause underestimation (by <2%) of the mass of some
factors, particularly HOA (hydrocarbon-like organic aerosol) and BBOA (biomass-burning organic aerosol). After





removing empty rows and columns, matrices from AMS$_{org}$ and AMS$_{org+inorg}$ analyses had the following dimensions: $193 \times 24{,}762$ and $205 \times 24{,}762$, respectively.

**2.3 Source apportionment: Positive matrix factorisation**

Source apportionment was performed using a PMF implemented in the multilinear engine (ME-2)(Paatero, 1997b, 1999). The PMF was configured and analysed using the SoFi (Source Finder) Pro interface (Canonaco et al., 2013)
(version 8.4.1.9.1; Igor 8.04). PMF is a descriptive mathematical algorithm that describes the input data, i.e. measurements of several variables collected over time (here, $m/z \times$ sampling time points), as a linear combination of factors that have constant mass spectra associated with temporally varying concentrations of the spectral constituents (Paatero, 1997a; Paatero and Tapper, 1994); each of the factors is representative of an emission source. The mathematical expressions and functions of the PMF algorithm have been exhaustively detailed in previous
studies (e.g., refs. (Tong et al., 2021; Stefenelli et al., 2019a; Chen et al., 2022; Chazeau et al., 2022), etc.).

We summarise the user-defined configurations applied in SoFi Pro to optimise the PMF of our datasets, PTR$_{CHARON}$, AMS$_{org}$, and AMS$_{org+inorg}$. The results were compared in terms of identified sources and the mass of OA (or total NR-PM$_1$) apportioned to each source.

**2.3.1. General methodology for PMF analysis**

Preliminary PMF was performed without using *a priori* information, i.e., the so-called unconstrained factorisation, to understand the data. These unconstrained trials explored solutions with three to 13 factors. Cell-wise, step-wise down-weighting was applied, whereby variables with S/N < 0.2 (bad variables) or 0.2 < S/N < 2 (weak variables) were down-weighted by a factor of 10 and 2, respectively (Paatero and Hopke, 2003; Ulbrich et al., 2009). Upon
establishing that primary factors, e.g., cooking and biomass burning, could readily be factorised in unconstrained trials, we explored only a subset of the possible solutions by directing the PMF toward meaningful solutions with the *a*-value approach. For this approach, the user can improve factorisation results by constraining the PMF with external data, if available (Canonaco et al., 2013; Paatero, 1999). For instance, a factor profile from a PMF trial in the same experiment, a time series from an external tracer from the same campaign, or a well-established factor
profile for a source from another experiment may be provided to the PMF as an 'anchor/vector' around which it can build a factor in its overall solution. The extent to which each PMF factor can diverge from the anchor is defined by the value of *a* (Tong et al., 2021), which varies from 0 to 1, where 0 = no divergence and 1 = up to 100% divergence. This anchor can be provided for one or multiple factors and has been proven to improve the quality of PMF solutions compared to unconstrained trials (Tong et al., 2021; Stefenelli et al., 2019a; Chen et al.,
250  2022).

Currently, there are no objective criteria for choosing the best number of factors in a solution; some criteria have been suggested in the literature to make an appropriate selection (Chen et al., 2022; Zhang et al., 2011; Ulbrich et al., 2009; Crippa et al., 2014). The PMF solutions reported here were primarily selected based on their
interpretability, which was in turn, determined by the distribution of known tracer compounds in the factors,




correlation with co-located measurements of external tracers (e.g., $NO_x$, $SO_2$), and the temporal agreement of factors determined by the two instruments. We resolved eight, four, and six factors from $PTR_{CHARON}$, $AMS_{org}$, and $AMS_{org+inorg}$, respectively. The justification for these choices is presented in **Table S2**. Once the most suitable solution, i.e., the base-case, was established, bootstrap analyses were performed to assess its stability, evaluate

uncertainties, and conduct a sensitivity analysis on the range of *a*-values used. In an unblocked bootstrapping approach, the original matrices (both data and error) are perturbed by random resampling of the rows to create a new input of the same dimensions, resulting in some duplications and deletions throughout the input (Paatero et al., 2014). The need and application of this approach differed between the $PTR_{CHARON}$ and the two AMS datasets as discussed in **Sections S5** and **S6**, respectively. Ancillary data on particle size distribution in each factor was

generated by a fully constrained PMF or simple linear regressions of the SMPS datasets (**Section S7**). Finally, the quality of solutions was gauged by the $Q/Q_{exp}$ values and from key diagnostic plots of residuals and the statistical stability across multiple runs (**Figure S5–S7**).

## 3 Results and Discussion

### 3.1 Campaign overview

**Figure 1** depicts a summary of the meteorological conditions, composition, and size distribution of NR-PM$_1$ observed from January 20 to February 26, 2022. Intense aerosol loads coincided with poor atmospheric dispersion due to slow wind speeds of less than 2 m/sec and strong surface-based temperature inversions (the difference in ambient air temperatures at 23 and 3 m was 3–10°C). The campaign-averages of BC and NR-PM$_1$ measured with the MAAP and AMS were **1.4 ± 1.4 µg/m³** and **8.3 ± 9.3 µg/m³**, respectively. Intense pollution episodes occurred

from Jan 31 to Feb 02, during which the daily average concentrations of NR-PM$_1$ were **24–27 µg/m³**. For this polluted period characterised by strong inversion, campaign-averaged PM$_{2.5}$ were ~25 and ~29 µg/m³ at NCore (a monitoring station located approximately 580 m from the CTC) using a beta attenuation mass monitor and a nearby site of Downtown using a DustTrak DRX aerosol monitor (Robinson et al., 2023). Conversely, the hourly NR-PM$_1$ concentrations measured at the CTC site comprised up to 99% of the PM$_{2.5}$ measured with an optical particle

counter, warranting that future studies in Fairbanks must explore the distribution, dynamics, and impacts of sub-micron aerosol to gauge the need for its targeted mitigation.

Organics were the predominant component of NR-PM$_1$ throughout the campaign, constituting **~66 ± 11**% of its total mass, while chloride, ammonium, nitrate, and sulphur-based inorganics contributed **2 ± 3, 3 ± 3, 6 ± 4**, and **22

± 10**%. This is in line with previous studies in Fairbanks, where OA was the largest component of PM$_{2.5}$ mass (Ward et al., 2012; Ye and Wang, 2020; Robinson et al., 2024). Specifically, according to a recent study from 2020 to 2021, ACSM analysis during the wintertime demonstrated inorganics to form less than 25% of the PM$_{2.5}$ mass only, with sulphate (~10%) and nitrate (~8%) being the predominant components (Robinson et al., 2024). Despite the different average concentrations, the fractional contributions of these non-refractory components remained

almost invariable throughout the campaign (**Figure 1D**). Detailed molecular-level composition of organics with the $PTR_{CHARON}$ reveals a large majority of organics to comprise only C, H, and/or O atoms, while only **~9 ± 4%** of





the OA$_{CHARON}$ mass measured with this instrument was attributable to heteroatomic species, including organonitrates and organosulphates (**Figure S8**). Generally, heteroatomic species cannot be distinguished at a resolving power of 5000 FWHM in complex environmental mixtures, such as atmospheric aerosol (Reemtsma, 2009). In this study, based on the low formula error and lack of an appropriate alternate, we gave 53 low-concentration ions (< 2% of the total signal) CHOS or CHNO identities, but due to the low confidence in their formula assignments, they were not considered for factor identification. Prominent peaks include *m/z* 217.09 ($C_{12}H_{12}N_2O_2$), 219.09 ($C_{15}H_{10}N_2$), 123.05 ($C_4H_{10}O_2S$), and 151.08 ($C_6H_{14}O_2S$).

On average, the OA mass loading recovered by PTR$_{CHARON}$ (i.e., OA$_{CHARON}$) accounted for approximately 85% of the OA mass measured by the AMS (i.e., OA$_{AMS}$). While the two instruments showed a strong temporal agreement ($R^2 = 0.60$) as depicted in **Figures 2A–B**, measurements were biased either toward the AMS$_{org}$ or the PTR$_{CHARON}$ (i.e., distributed away from the 1:1 line in the scatter plot of **Figure 2C**) during different periods of the campaign. These trends could unequivocally be explained by the variation in relative contributions of two major emission sources identified by both instruments in this study: on-road traffic and biomass burning. OA$_{CHARON}$ was comparable to OA$_{AMS}$, when the relative contribution of BBOA$_{AMS,org}$ was more than 50% of total OA$_{AMS}$ and HOA$_{AMS,org}$ (i.e., traffic$_{CHARON}$) was less than 10% (**Figure 2D–E**). Similar trends were observed for some major constituents of BBOA, e.g., levoglucosan and a PAH ($C_{20}H_{12}$) as shown in **Figure S9**. This relationship of instrument performance with the source can be traced back to the size of particles, where sub-100 nm urban vehicular emissions are underestimated by the PTR$_{CHARON}$ (Guo et al., 2020; Pikridas et al., 2015; Louis et al., 2017; Kostenidou et al., 2020), and larger than 100 nm biomass burning emissions (Reid et al., 2005) are estimated well (Janhäll et al., 2010).

Part of the quantitative difference between the two instruments can also be explained by the fragmentation of analyte ions during PTR ionisation that introduces a negative bias. This bias has been reported to be small for oxidised organic compounds (Leglise et al., 2019). Additional tests carried out in our laboratory with five $C_{16}$–$C_{26}$ alkanes as markers of vehicle emissions revealed that fragmentation increases dramatically and results in a 2–4 times underestimation of actual concentrations. The tendency of alkanes from vehicular exhausts to undergo dissociative PTR ionisation has also been reported previously (Gueneron et al., 2015).

## 3.2 Source apportionment

### 3.2.1. Overview of source apportionment

A four-factor solution was selected for the AMS$_{org}$ measurements with three primary factors (i.e., HOA, COA, and BBOA) and an oxygenated or aged OA factor (i.e., OOA). The mass spectra and time series are presented in the supplement **(Figure S10)**. Counterparts of these four factors were diagnosed in AMS$_{org+inorg}$ based on a high temporal correlation ($R^2 > 0.9$; **Table S4**), along with two additional factors: a sulphur-rich factor (labelled sulph-OA) and a nitrate-rich factor (labelled AmNi) (**Figure 3**). An eight-factor solution was selected for PTR$_{CHARON}$ and is summarised in **Figures 4** and **5**. To differentiate between corresponding factors retrieved from the different





datasets, they have been given unique subscripts, e.g. $COA_{AMS,org}$, $COA_{AMS,org+inorg}$, $COA_{AMS}$ (i.e., referring to both AMS datasets), or $COA_{CHARON}$. Amongst the three datasets COA, HOA (labelled 'traffic' in $PTR_{CHARON}$ analyses), and OOA were common. A single BBOA factor was observed in $AMS_{org}$ and $AMS_{org+inorg}$, while four chemically distinct, but closely co-varying counterparts were detected by $PTR_{CHARON}$.

### 3.2.2. Organic aerosol from residential heating

Both AMS analyses indicate that biomass burning is among the major sources of $PM_1$ during the ALPACA campaign. On average, BBOA contributed **1.5 ± 1.9 μg/m³** (**28 ± 18%** of total $OA_{AMS}$) and **1.6 ± 2.2 μg/m³** NR-$PM_1$ (**19 ± 14%** of total NR-$PM_1$ mass). The mass spectra of $BBOA_{AMS}$ featured a strong peak at $m/z$ 60 ($C_2H_4O_2^+$) and 73 ($C_3H_5O_2^+$)(**Figure S10A–B**). These fragments are markers of anhydrosugars in wood-forming polymers, such as cellulose (Tobler et al., 2021). Wood combustion has previously been estimated to be the largest emitter of aerosols in Fairbanks and surrounding areas, where it may produce as much as 80% of the aerosol load (Haque et al., 2021; Ward et al., 2012; Wang and Hopke, 2014; Kotchenruther, 2016). Wood burning emissions are also the major driver of the spatial variability of $PM_{2.5}$ and BC in Fairbanks during strong atmospheric temperature inversions (Robinson et al., 2023). Other typical residential heating sources of emissions in Fairbanks include coal, gas, and fuel oil (Simpson et al., 2019).

The $BBOA_{AMS}$ factor was strongly correlated with PAHs ($R^2 \geq 0.7$). In addition, a moderate correlation was observed with $SO_2$ ($R^2 = 0.4$) (**Table 1**). While PAHs are a major component of biomass combustion emissions, the emission of $SO_2$ is largely associated with coal and oil combustion (Smith et al., 2011; Dunleavy and Brune, 2019). However, the AMS was unable to distinguish between multiple combustion-related sources. As shown in the diurnal plots in **Figure 3**, the concentration of the $BBOA_{AMS}$ factor enhanced at ~1800 AKST, stayed stable through the night and then decreased in the early morning. Its lowest mass concentrations occurred during the afternoon (1300–1500 AKST). Therefore, $BBOA_{AMS}$ could be associated with residential heating, i.e., the combustion of a variety of fuels by residents within their homes (non-commercially), such as in wood-burning stoves, furnaces, boilers, etc. for heating living space. We did not find evidence of OA or NR-$PM_1$ from commercial heat providers, such as power plants, likely due to their small contribution to surface-level aerosol due to smokestacks lying above the inversion layer.

$PTR_{CHARON}$ apportioned **2.6 ± 3.4** μg/m³ of $OA_{CHARON}$, on average, to four distinct residential heating-related sources expressed as ResH1–4 (**62 ± 26%** of total $OA_{CHARON}$). These factors closely co-varied in time and were correlated reasonably well or strongly ($R^2 = 0.5$–$0.7$; **Table S5**) with the $BBOA_{AMS}$ factors. In addition, combining all four residential heating-related factors in $PTR_{CHARON}$ into a composite factor increased the correlation ($R^2$) with $AMS_{org}$ and $AMS_{org+inorg}$ to 0.79 and 0.82, respectively, suggesting that PMF was not able to effectively separate these closely co-varying residential heating factors when their molecular signatures were weakened due to the extensive EI-induced fragmentation in AMS. The four factors from $PTR_{CHARON}$ were identified as different sources based on the distribution of key marker species and correlation with external (e.g., trace gases, etc.) and internal (e.g., PAHs measured with co-located instruments; particle size distribution) tracers. The levoglucosan ion is used




here as an internal tracer of biomass burning because it is relatively stable under typical atmospheric conditions (Fraser and Lakshmanan, 2000). A majority of the signal from protonated levoglucosan (*m/z* 163) and its fragments (at *m/z* 85, 127, and 145) appeared in ResH1, ResH4, and ResH2 (in the same order), with only minor association
with ResH3, suggesting the former three to originate from biomass burning – more specifically, wood-burning (**Figure 4 and S11**). These three wood-burning related factors collectively produced an average **of 2.1 ± 2.5 μg/m³** of OA$_{CHARON}$ (**47 ± 20%** of total factorised OA$_{CHARON}$).

*ResH1 includes mixed wood-burning OA:* Approximately, 30, 14, 9, and 26% of the protonated levoglucosan
signal was distributed in ResH1 to ResH4 respectively, with a similar trend for the fragments. Although ResH1 had the strongest levoglucosan signal, it contributed the least OA with an average of **0.5 ± 0.5 μg/m³** and did not feature any other prominent wood-burning tracers, such as PAHs. As shown in **Figure S13**, ~65% of the total signal of ResH1 came from compounds with six or fewer carbon atoms, compared to heavier species in other factors. Many species with the greatest concentrations in ResH1, relative to other factors, have been reported as oxidation products
of BBOA ageing in previous studies, such as *m/z* 69.03 (C$_4$H$_4$O; furan) (Palm et al., 2020; Jiang et al., 2019), *m/z* 87.04 (C$_4$H$_6$O$_2$; oxobutanal) (Brégonzio-Rozier et al., 2015), *m/z* 97.03 (C$_5$H$_4$O$_2$; furfural), and *m/z* 109.0286 (C$_6$H$_4$O$_2$; benzoquinone)(Stefenelli et al., 2019b). Consistent with this, the concentration-weighted average O/C of ResH1 was relatively higher (i.e., 0.42) compared to other residential heating factors (O/C = 0.2–0.3). Collectively, ResH1 comprises OA from the combustion of a variety of mixed wood-based solid fuels as evidenced by the
presence of levoglucosan, but it also likely includes OA in the early stages of processing.

*ResH2 and ResH4 include OA from hardwood and pinewood combustion, respectively:* Two more factors associated with wood-burning were ResH2 and ResH4. Their average OA$_{CHARON}$ concentrations were **1.1 ± 1.9** and **0.8 ± 0.9 μg/m³**, respectively (**Figure 6**). As shown in **Figure 6A**, ResH2 was the single most dominant factor in
the PMF of PTR$_{CHARON}$ that contributed up to **~37 μg/m³** of OA$_{CHARON}$ alone during the most severe pollution episodes. Not only did these factors correspond to OA particle sizes greater than 300 nm (**Figure S12**), which is typical of woodsmoke (Glasius et al., 2006), but they also presented unique molecular signatures of different wood types as shown in **Figure S11** and discussed next. Generally, the specific nature of wood cannot be inferred unambiguously because the emissions of known marker species, such as levoglucosan or methoxy phenols, vary
not just with fuel used and its quality, but also with the type of heating appliance, operational conditions, appliance efficiency, and stage in the combustion cycle (Fine et al., 2002; Alves et al., 2017). Regardless, several studies have distinguished between softwood from hardwood by investigating the presence of marker compounds that were observed in our study as well, such as substituted phenols and resin acids.

ResH2 featured an abundance of prominent methoxy phenols, including C$_7$H$_8$O$_2$ (guaiacol), C$_8$H$_{10}$O$_3$ (syringol), C$_{10}$H$_{10}$O$_3$ (coniferaldehyde), C$_6$H$_6$O$_2$ (benzenediol (catechol) or methylfurfural), and C$_8$H$_{10}$O$_2$ (creosol), where they collectively accounted for ~9% of the total signal, compared to 1, 2, and 2% in ResH1, ResH3, and ResH4, respectively. These compounds are important products of lignin pyrolysis in birch, aspen, and spruce and are usually found in the gas phase at mild ambient temperatures (Kong et al., 2021). Guaiacol and syringol are
depolymerisation products of guaiacyl and syringyl units of lignin at 200–400°C, and they rapidly transition to





catechols, cresols, and phenols during secondary pyrolysis reactions at 400–450°C, eventually leading to enhanced PAH formation at >700°C (Kawamoto, 2017). While guaiacols are emitted to some extent by the burning of both hardwood and softwood, semi- or low-volatility substituted syringols that primarily exist in the condensed phase are emitted in much higher amounts by hardwood combustion (Kawamoto, 2017; Fine et al., 2002, 2001; Schauer and Cass, 2000). In this study, derivatives of guaiacols, including $C_{10}H_{12}O_2$ (eugenol), $C_{10}H_{14}O_2$ (4-propyl guaiacol), and $C_{10}H_{10}O_3$ (coniferaldehyde), presented much higher relative concentrations (i.e., 'normalised concentration of a variable in a given factor' - 'average normalised concentration of variable across all factors' / 'standard deviation of its concentration across all factors') of 0.56–1.41 for ResH2 and ResH4 compared to < 0 for ResH1. Other compounds, such as $C_8H_8O_3$ (vanillin), $C_9H_{10}O_3$ (acetovanillone), $C_{10}H_{12}O_3$ (propiovanillone), and $C_{10}H_{12}O_4$ (methyl homovanillate), were predominantly found in ResH2. Similarly, substituted syringols, i.e., $C_{11}H_{14}O_3$ (methoxy eugenol), $C_{10}H_{12}O_4$ (acetosyringone), and $C_{11}H_{14}O_4$ (syringyl acetone, propionyl syringol, or sinapyl alcohol) were almost entirely associated with ResH2 as well. These compounds have been reported as markers of hardwood burning (Fine et al., 2001), implying a potentially greater contribution of hardwood smoke to the ResH2 factor. In Alaska, relevant hardwood species include deciduous leafy trees, i.e., paper birch, balsam poplar, quaking aspen, etc (ADEC, 2023).

For ResH4, in addition to the levoglucosan marker ions, a predominance of large, oxygenated molecules with more than 13 carbon atoms was observed (**Figure S13**), such as $C_{16}H_{30}O_6$ (*m/z* 319.21), $C_{20}H_{28}O_2$ (*m/z* 301.21), $C_{22}H_{18}O$ (*m/z* 299.14), $C_{20}H_{18}O_4$ (*m/z* 323.12), and $C_{20}H_{30}O_2$ (*m/z* 303.24). Amongst these, more than 60% of the signal from *m/z* 301 ($C_{20}H_{28}O_2$) and *m/z* 303 ($C_{20}H_{30}O_2$) was associated with ResH4 (**Figure S11**). These species are likely resin acids, dehydroabietic acid and abietic acid, respectively, which are almost exclusively emitted from the thermal alteration of resins in coniferous species, and thus, are indicative of softwood burning (Simoneit, 2002, 1999). Owing to the presence of these compounds, ResH4 was identified as OA influenced by softwood combustion. Softwood species in Alaska include trees with needles and cones, e.g. hemlock, cedar, and spruce (ADEC, 2023).

***ResH3 includes OA from heating oil combustion:*** A factor, labelled ResH3, contributed **16 ± 9%** of the total $OA_{CHARON}$ (**0.6 ± 0.6 µg/m³**) and showed the characteristic diurnal pattern of residential heating. It correlated well ($R^2$ = 0.56) with $BBOA_{AMS,org}$. However, its chemical composition was very different from the other residential heating factors. Notably, levoglucosan contributed to a smaller fraction of the total signal of ResH3 (i.e., 9%) compared to other residential heating factors (13–29%; **Figure S11**), but PAHs and condensed aromatic species represented a much larger fraction of its total signal (for instance, 30, 31, and 29% of $C_{16}H_{10}$ (*m/z* 203.09), $C_{18}H_{12}$ (*m/z* 229.10), and $C_{20}H_{12}$ (*m/z* 253.10) compared to 0–18, 0–21%, 0–17% for ResH1–2, and ResH4; Figure S13). These PAHs could be fluoranthene (or pyrene), naphthacene (or benzo[*x*]anthracene, chrysene), and benzo(*x*)pyrene (or benzo(*x*)fluoranthene)), which have been reported in emissions of light oil combustion (Bari et al., 2009). Additionally, ResH3 was strongly correlated with $SO_2$ ($R_2$ = 0.61) during the campaign, compared to a moderate correlation of ≤ 0.47 with the remaining residential heating factors. Residential combustion of heating oil is an important source of $SO_2$ in Fairbanks, compared to wood and coal, due to ~2/3$^{rd}$ of the households using oil-fired space heaters and the high sulphur content of > 1600 ppm in fuel oils commonly consumed here (e.g., #1 and




#2 fuel oil and waste motor oil are relevant in Fairbanks)(Dunleavy and Brune, 2019). Consistent with the possibility of the ResH3 factor denoting fuel oil emissions, a fully constrained PMF on SMPS measurements matched it with particles smaller than 100 nm (Figure S12). Due to the small particle size, it is possible that mass concentrations of $OA_{CHARON}$ were under-apportioned to ResH3; this possibility is discussed in detail for the on-road traffic factor in the next section.

**3.2.3. Hydrocarbon-like and cooking organic aerosol**

The $HOA_{AMS}$ factors were characterised by notable peaks at $m/z$ 43 ($C_3H_7^+$), 57 ($C_4H_9^+$), 71 ($C_5H_{11}^+$), 85 ($C_6H_{13}^+$), and 99 ($C_7H_{15}^+$) belonging to $[C_nH_{2n+1}]^+$ series that are typical of n- and branched alkanes. There were also $m/z$ 55 ($C_4H_7^+$), 69 ($C_5H_9^+$), 81 ($C_6H_9^+$), 83 ($C_6H_{11}^+$), 95 ($C_7H_{11}^+$), 97 ($C_7H_{13}^+$), 107 ($C_8H_{11}^+$), 109 ($C_8H_{13}^+$), and 111 ($C_8H_{15}^+$) that belong to $[C_nH_{2n-1}]^+$ and $[C_nH_{2n-3}]^+$ series, which are typical of cycloalkanes. These are key ions associated with
engine-lubricating oils, vehicular exhaust, and diesel fuel (Canagaratna et al., 2004). The $HOA_{AMS,org}$ and $HOA_{AMS,org+inorg}$ factors contributed **38 ± 20%** (of the $OA_{AMS}$) and **21 ± 14%** (of the total NR-PM$_1$) mass, respectively (**Figures 6 and S14**). HOA is generally associated with vehicular emissions from on-road traffic, which were not observed in the unconstrained PMF of $PTR_{CHARON}$. However, a factor for on-road traffic was 'artificially' diagnosed in the $PTR_{CHARON}$ analysis by constraining the factorisation with the time series of a mobile
gasoline factor identified in the gas-phase PTR-ToF MS analyses in the ALPACA campaign (Temime Roussel et al., 2022). The success of constraining this factor was evident in characteristics typical of on-road traffic. For instance, it was strongly correlated with black carbon and NO$_x$ (R$^2$ of 0.58 and 0.66; **Table 1**) and featured high contributions of $C_8H_{10}$ (xylene; ethylbenzene; 2%), $C_7H_8$ (toluene; 4%), and $C_6H_6$ (benzene; 0.5%) to its total mass concentrations (**Figures 4 and S11**). In addition, peaks in the daily average mass concentrations of the traffic$_{CHARON}$
factor coincided with morning (0900 AKST) and evening (1700–1600 AKST) rush hours (**Figure 5**). However, the traffic$_{CHARON}$ factor had negligible concentrations (< 1 µg/m$^3$) and contained implausible species, such as $m/z$ 315.22 ($C_{21}H_{30}O_2$; possibly cannabidiol) that would otherwise (e.g., in unconstrained trials) appear as PMF residuals, making its environmental representativeness suspicious.

Another primary factor identified in Fairbanks was cooking, which could either be from residential or commercial activities around the CTC. Both $COA_{AMS}$ factors featured a high abundance of $C_xH_y^+$ ions, along with prominent O$_1$ fragments at $m/z$ 55 ($C_3H_3O^+$), 84 ($C_5H_8O^+$), and 98 ($C_6H_{10}O^+$), which originate from organic acids (Mohr et al., 2009). These fragments have been reported as diagnostic spectral markers of COA in urban settings (Sun et al., 2011). The $f$55/$f$57 value (i.e., the ratio of fractions of $C_4H_7^+$ to $C_4H_9^+$) was ~3.00 for $COA_{AMS}$, compared to ~1.04
in $HOA_{AMS}$ (**Figure S10D**). A high $f$55/$f$57 ratio of >1 is considered a characteristic feature of COA (Katz et al., 2021; Sun et al., 2011) because a reliable external tracer for it is yet to be identified. The PMF analysis of $PTR_{CHARON}$ also contained a distinct COA factor dominated by long-chain fatty acids, $C_{18}H_{32}O_2$, $C_{18}H_{34}O_2$, and $C_{18}H_{36}O_2$, identified here as linoleic, oleic, and stearic acids that contributed 11, 16, and 4% to the total $COA_{CHARON}$ mass (**Figure 4 and S11**). These fatty acids are common markers of OA from cooking oil and meat (Katz et al.,
2021; Mohr et al., 2009). Across the whole campaign, $COA_{CHARON}$ made its highest contributions of ~9% to the total $OA_{CHARON}$ mass a little after noon (lunchtime) and in the evening (dinnertime) resulting in the unique diurnal pattern visualised in **Figure 5**.





Quantitatively, there was a large discrepancy between the OA apportioned to HOA and COA by PTR$_{CHARON}$ and AMS$_{org}$. For instance, on average, **2.1 ± 3.0 µg/m³** of OA was associated with HOA$_{AMS,org}$ during the campaign, compared to only **0.1 ± 0.1 µg/m³** for the traffic$_{CHARON}$ factor (**Figure 6**). Similarly, average absolute concentrations of COA$_{AMS,org}$ and COA$_{CHARON}$ were **0.6 ± 0.8** and **0.1 ± 0.2 µg/m³**, respectively (**Figure 6**). We speculate that the shortcomings seen in OA mass measured by the PTR$_{CHARON}$ relative to the AMS$_{org}$ were largely instrumental, such as the low sensitivities of the PTR$_{CHARON}$ for small particles (<100 nm) and hydrocarbons. Previous studies using the PTR$_{CHARON}$ in Innsbruck, Austria, successfully observed a traffic factor, but no cooking emissions despite sampling at an urban locality (Müller et al., 2017). A variety of environmental and user biases could also be involved, such as the contribution of non-vehicular sources to the HOA$_{AMS}$ factors and the choice of suboptimal conversion coefficients (e.g., RIE) in the AMS analyses (see **Sections S8** and **S9** for details). These are important considerations in employing the PTR$_{CHARON}$ for ambient air analyses because a full picture of the sources involved, especially in urban regions influenced by primary OA emissions of smaller particle sizes, may not be possible without complementary measurements.

### 3.2.4. Oxygenated organic aerosol

It is common in past source apportionment studies to report multiple OOA factors that differ in volatilities or oxygenation levels (e.g., (Stefenelli et al., 2019a; Kumar et al., 2022; Cash et al., 2020)), but we diagnosed only a single OOA factor in either AMS or PTR$_{CHARON}$ measurements. The OOA$_{AMS,org}$ factor was identified based on a prominent peak at $m/z$ 43 ($C_2H_3O^+$), which is a tracer of less oxygenated OA, and $m/z$ 29 ($CHO^+$; **Figure S10A**). It correlated strongly with OOA$_{CHARON}$ with an $R^2$ of 0.74, where the average absolute concentrations of OOA$_{CHARON}$ and OOA$_{AMS,org}$ were **0.4 ± 0.6** and **1.0 ± 2.1 µg/m³**. Some of the most intense ions in the mass spectra of OOA$_{CHARON}$, relative to other factors, included $m/z$ 99.01 ($C_4H_2O_3$, e.g., maleic anhydride), $m/z$ 167.10 ($C_{10}H_{14}O_2$), $m/z$ 127.08 ($C_7H_{10}O_2$; e.g., heptadienoic acid), $m/z$ 185.10 ($C_{13}H_{12}O$; e.g., benzyl phenol), and $m/z$ 171.07 ($C_8H_{10}O_4$), as well as some species that overlapped with the residential heating factors, notably $m/z$ 163.06 ($C_6H_{10}O_5$; e.g., levoglucosan), $m/z$ 179.08 ($C_{10}H_{10}O_3$; e.g., coniferaldehyde), and $m/z$ 301.21 ($C_{20}H_{28}O_2$; e.g., dehydroabietic acid). Some of these species (e.g., $C_4H_2O_3$, $C_{10}H_{14}O_2$, $C_7H_{10}O_2$) have previously been associated with atmospheric oxidation or photolysis of BBOA (Montoya-Aguilera et al., 2017; Lignell et al., 2013; Smith et al., 2020).

Given the prominence of wood-burning as the major source of primary emissions in ALPACA, the OOA is likely linked to BBOA. A recent source apportionment of NR-PM$_1$ measured with the HR-ToF AMS at a site close to the CTC did not reveal an OOA factor at all, while BBOA, HOA, and mixed primary factor (HOA, COA, etc.) comprised 45, 25, and 31% of total OA, on average, during the campaign (Yang et al., 2024). Minimal processing, and thus, limited OOA formation is plausible due to short solar light exposure periods and pollution residence in Fairbanks (Cesler-Maloney et al., 2024), but a complete disappearance of OOA is more likely to be a consequence of it remaining unresolved under the factorisation method used. Another recent study in Fairbanks using the ACSM identified wintertime OOA as a mixture of real BBOA and SOA formed from non-photochemical processing (Robinson et al., 2024). This aspect was investigated via an $f$44 versus $f$60 plot for AMS$_{org}$ that supports some





influence of biomass burning at all levels of oxidation of OA (**Figure S10C**). The placement of OOA$_{AMS,org}$ toward the left edge of the $f44$ versus $f60$ plot is consistent with aged OA from wood burning (Xu et al., 2023), but an urban influence cannot be ruled out in field settings (Cubison et al., 2011), especially when $m/z$ 60 and 73 are only 0.2 and 0.4% of the total OOA$_{AMS,org}$ signal (**Figure S10B**).

525

Much more interesting information regarding the OOA factor was gleaned from the AMS$_{org+inorg}$ measurements, which revealed it to be rich in sulphur (**Figure S15**). The AMS does not quantitatively distinguish among the different sulphur-containing species, such as hydroxymethane sulphonate (HMS; $CH_2(OH)SO^{-3}$), $SO_3^{2-}$ (sulphite), $HSO_3^-$ (bisulphite), and $SO_4^{2-}$ (sulphate), or between organic and inorganic sulphur. Therefore, we used the ratio of these fragments to speculate on the different forms. This was inspired by previous studies on sulphur source apportionment with the AMS and fragmentation patterns (Chen et al., 2019; Schueneman et al., 2021), whereby we performed calibrations on the AMS with pure $(NH_4)_2SO_4$ mixed with various amounts of levoglucosan (i.e., 0–80% in mass). This mixture was used to mimic the matrix effect that can potentially impact sulphur fragmentation patterns by wood smoke as previously demonstrated by Schueneman et al., 2021. We compared the fractions of $HSO_3^+$ to $H_2SO_4^+$ fragments normalised to the fractions $H_2SO_4^+$ and $HSO_3^+$ for pure $(NH_4)_2SO_4$ (Chen et al., 2019). Results are shown in **Figure S16A**, where the OOA$_{AMS,org+inorg}$ factor exhibited much lower $HSO_3^+$ to $H_2SO_4^+$ intensities, which is indicative of an organosulphur influence.

Organosulphur content was thus calculated using the ratios of $SO^+$ and $SO_2^+$ ions against $SO_3^+$, $HSO_3^+$, and $H_2SO_4^+$ ions in the AMS spectra, as detailed by (Song et al., 2019). It constituted as much as $20 \pm 16\%$ ($0.8 \pm 1.3$ µg/m$^3$) of all sulphur measured by the AMS, which increased to $23 \pm 12\%$ ($0.9 \pm 0.6$ µg/m$^3$) during a pollution period (Jan 30–Feb 02, 2022); this is consistent with previous reports on organosulphur being a substantial component of particulate sulphur during pollution events (Campbell et al., 2022; Robinson et al., 2024). In line with the $f$HSO$_3$/$f$H$_2$SO$_4$ analysis shown in **Figure S16A**, the estimated organosulphur fraction was mainly associated with the OOA$_{AMS,org+inorg}$ factor (R$^2$ = 0.85) (**Figure S16D–E**). The total concentration of sulphur-related fragments in OOA$_{AMS,org+inorg}$ was **$0.9 \pm 1.8$ µg/m$^3$**, on average, and accounted for **$26 \pm 23\%$** of the total sulphur measured with the AMS, which agrees with the theoretical estimation of organosulphur content (Song et al., 2019). Further information on chemical composition was gathered by comparing this factor with IC measurements from PM$_{0.7}$ filter samples analysed as part of another ALPACA study (Dingilian et al., 2024). Both methods (IC and AMS) correlated well, despite a negative bias against the AMS analysis that underestimated the sums of sulphur-, ammonium-, and nitrate-related fragments (see **Section 2.2.2** for fragments included) by ~ 31, 26%, and 35% compared to the IC analyses (**Figure 7A**). Both the total estimated organosulphur and OOA$_{AMS,org+inorg}$ factor presented very strong correlations (R$^2$ > 0.90) with the S$_{(IV)}$ and HMS ions (**Figures 7B and S16F–I**) with a relatively weaker, but still strong correlation (R$^2$ > 0.61–0.68) with the SO$_4^{2-}$ ion.


S$_{(IV)}$ species, including HMS, have been observed as the major secondary organosulphur component of PM$_{2.5}$ in Fairbanks during wintertime with average concentrations of 0.29 and 0.34 µg/m$^3$ recorded with IC in 2020 and 2021, respectively, contributing 26–41% of total sulphur (Campbell et al., 2022). Recently, co-varying HMS and S$_{(IV)}$ species were distinguished in Fairbanks, and the non-HMS S$_{(IV)}$ were reported to be aldehyde-S$_{(IV)}$ compounds



(Dingilian et al., 2024). In addition, this factor was very strongly correlated with total ammonium ($R^2 = 0.95$, **Table 1**; **Figure S16D–E**) which could raise aerosol pH, favouring the formation of $S_{(IV)}$ species under appropriate meteorological conditions and aerosol composition (Campbell et al., 2024). Therefore, the presence of HMS and other organic $S_{(IV)}$ species in the $AMS_{org+inorg}$ factor is well-substantiated. Overall, based on the molecular composition from $PTR_{CHARON}$ and chemical information from $AMS_{org+inorg}$, as well as the diurnal pattern with peak

concentration in the afternoon (**Figure 3**) that is indicative of chemical daytime processing, the wintertime OOA in Fairbanks is not solely HMS; it is instead a mixture of secondary non-heteroatomic organic matter and organosulphur compounds, which hints toward its formation from complex atmospheric processing pathways that needs further exploration.

**3.2.5. Additional insights from combined analysis of organic and inorganics in AMS measurements**

Two additional factors, sulph-OA (i.e., sulphur-rich OA) and AmNi (i.e., ammonium nitrate), were exclusively observed from the PMF of $AMS_{org+inorg}$. Approximately 40–60% of these factor's masses comprised sulphur and nitrogen species (**Figure S15**).


***Sulphur-rich organic aerosol:*** Like the $OOA_{AMS,org+inorg}$ factor, sulph-OA was also sulphur-rich. Its chemical composition was explored via the $fHSO_3/fH_2SO_4$ analysis detailed in **Section 3.2.4**. This factor lay in the upper-right quadrant of **Figure S16A**, where it was aligned between pure $H_2SO_4$ and/or $(NH_4)_2SO_4$. The measured $[NH_4]/[SO_4]$ ratio for sulph-OA was 0.07, which is much lower than the theoretical mass ratio of 0.38 and 0.18 for

$(NH_4)_2SO_4$ and $NH_4HSO_4$, respectively. Therefore, this factor is inferred to have an acidic nature.

Notably, the sulph-OA factor was strongly correlated with $SO_2$ ($R^2 = 0.6$), which is majorly a primary product of residential heating oil (Dunleavy and Brune, 2020). Therefore, it is likely that sulph-OA comprises primary ultrafine emissions in the range of 50–80 nm from heating oil combustion (**Figure S12D**). This factor contained

**0.6 ± 0.5 µg/m³** of sulphur. Despite the low concentrations, sulph-OA made up **58 ± 26%** of total sulphur measured with the AMS because it dominated during the low-pollution periods, which were more frequent and lasted longer than the high-pollution periods (**Figure 1**). Other primary factors, $HOA_{AMS,org+inorg}$, $COA_{AMS,org+inorg}$, and $BBOA_{AMS,org+inorg}$, contained an additional **11 ± 9%** of the sulphur (**0.2 ± 0.2 µg/m³**, on average). Primary sulphur factors collectively made up **69 ± 24%** (**0.7 ± 0.6 µg/m³**) of total sulphur. This value is in close agreement with a

previous ALPACA study that reported ~62 ± 12% of total $SO_4^{2-}$ mass to be primary and associated with particles of smaller than 700 nm ($2.1 ± 1.4$ µg/m³ in $PM_{0.7}$) (Moon et al., 2023).

Surprisingly, sulph-OA was only moderately correlated with the ResH3 factor ($R^2$ of 0.33), which was identified as heating oil OA in the $PTR_{CHARON}$ analysis. Specifically, the sulph-OA factor made relatively higher contributions

to $NR-PM_1$ and correlated with $SO_2$ only during low-pollution episodes, when the contributions and absolute concentrations of all other factors (including ResH3) decreased. Regardless of the low correlation, we speculate that ResH3 and sulph-OA represent the same source, i.e., residential heating fuel combustion, and their temporal disagreement may result from instrumental biases in quantifying particles smaller than 100 nm (**Figures S12B and**





**D**). For instance, as shown in **Figures S12E–F**, the organic-only ResH3 supersedes sulph-OA concentrations, when
larger particles are abundant, and it has lower concentrations for smaller particles.

*The AmNi factor includes atmospherically processed vehicular emissions:* The second inorganic factor was
composed of 35% nitrates, 14% ammonium, and 43% organics. It accounted for $71 \pm 23\%$ of the total nitrate
measured in NR-PM$_1$ ($R^2 = 0.98$). The average concentrations of this factor and the nitrate species in it were **1.1 ±
1.6 μg/m$^3$ and 0.4 ± 0.5 μg/m$^3$**. It presented a distinct peak from ~1200–1800 hrs and then stable, low
concentrations throughout the night (**Figure 3**). This peak followed 3–4 hours after the peak in the mass
concentrations of HOA$_{AMS}$ (or traffic$_{CHARON}$) during the morning, implying its probable origin from vehicular NO$_x$,
which was supported by the highest contributions of this factor coinciding with peaks in NO$_x$ concentrations
(**Figure S17B**). Generally, during the ALPACA campaign, the AmNi factor had much lower concentrations than
HOA$_{AMS,org+inorg}$; however, they were both associated with the highest recorded ambient temperatures (5 to -10°C)
and solar radiations (as per $j$NO$_2$ values)(**Figure S17C–D**). According to atmospheric modelling studies in
Fairbanks (Joyce et al., 2014), the formation of NO$_3$ from NO$_x$ via the nocturnal reactions slows at temperatures
below -15°C, causing them to have higher concentrations during warmer periods.

Interestingly, according to the difference in mass concentrations of HOA$_{AMS,org}$ and HOA$_{AMS,org+inorg}$ and its
correlation with the AmNi factor (**Figure S17A**), we speculate that some portion of the organic components of the
AmNi factor were apportioned to HOA$_{AMS,org}$, causing it to have higher contributions than HOA$_{AMS,org+inorg}$ (**Figure
6**). The inclusion of inorganics provided more variables to the PMF, and thus, improved the resolution of factors
into distinct AmNi and HOA$_{AMS,org+inorg}$ factors.

## 4. Local environmental implications and conclusive remarks

We surmise from PTR$_{CHARON}$ and AMS analyses that primary emissions from residential heating and on-road traffic
are collectively responsible for producing more than half of the sub-micron aerosol mass in Fairbanks during the
wintertime. We show that PTR$_{CHARON}$ helped resolve residential heating OA into four distinct sources based on
hardwood, softwood, and fuel oil combustion, while AMS$_{org}$ analysis yielded a single composite BBOA factor.
This enhanced deconvolution and quantification of closely co-varying sources of ambient pollution epitomises the
novelty of our study and has implications for the development of air quality regulation and allows gauging public
adherence to it.

For instance, during this study, 12–48-hour-long ADEC advisories for wood-burning restrictions were
implemented seven times. Variation in the relative contributions of ResH1–4 during these advisories is depicted in
**Figures 8** and **S18–21**. For all advisory events, ResH2 and ResH4, i.e., woodsmoke, were the predominant
contributors *before* and *after* the advisories were in place. ResH2 (i.e., hardwood-related fuels) remained a
prominent contributor to OA$_{CHARON}$ *during* the 3$^{rd}$ (Stage 2), 4$^{th}$ (Stage 1), and 5$^{th}$ (Stage 1) advisories. A notable
increase was observed in ResH3 contribution, i.e., heating oil, at least once *during* the 2$^{nd}$ (Stage 1), 5$^{th}$ (Stage 1),
6$^{th}$ (Stage 1), and 7$^{th}$ advisory events. Most households in Fairbanks use heating oil (~72% of residents), followed



by wood (~22% of residents) (Dunleavy and Brune, 2019), which was not reflected here proportionately in the relative contributions of ResH3. This can be linked to a higher $PM_1$ release from wood combustion per given volume of fuel compared to other commonly used sources, including heating oil, especially under less-than-optimal combustion conditions (e.g., moist wood) or with inefficient appliances. There is also the possibility that due to the

typical particle size of ResH3 emissions being smaller than 100 nm (**Figure S12**), this source was not efficiently quantified by the $PTR_{CHARON}$.

All seven ADEC advisories coincided with the coldest periods of the campaign (**Figure 1**). Therefore, the response of Fairbanks' residents to ADEC advisories cannot be assessed independently from their response to increased need

for heating or the dynamics of OA under the unique meteorology (i.e., low temperatures/low solar radiations/strong inversions) during sampling. In our study, the absolute average concentrations of all factors were inversely related to ambient temperature, but the percent change differed considerably across factors. Specifically, as temperatures decreased from -10°C to below -25°C, the average absolute concentrations for $traffic_{CHARON}$, $COA_{CHARON}$, $OOA_{CHARON}$, ResH1–4 increased 0.25×, 0.75×, 9.0×, 1.4×, 25.1×, 3.0×, and 2.9×, respectively (**Figure S22**). The

steep increase in the relative contribution of ResH2 was associated with hardwood-based fuels. In contrast, based on surveys (Dunleavy and Brune, 2019) and ratios of organic tracers in ambient air samples (Haque et al., 2021), previous studies reported birch and spruce, which are widely found in Alaskan boreal forests, as the most popular firewood in Fairbanks during winters. Laboratory studies have shown that the burning of softwood pellets of Douglas Fir or eastern white pine emits less PM than hardwood pellets of the same volume, and this response varies

based on the moisture content of the wood and the heating appliance used (Morin et al., 2022). High PM emission per volume burned could also be the reason behind hardwood burning being the dominant contributor of PM in our analysis. ResH2 comprises a broader spectrum of volatile and semi-volatile substituted phenolic species, and thus, it is likely to undergo gas-to-particle partitioning at low temperatures toward increasing OA loads (Ijaz et al., 2025).

Overall, investigating the variation in the emission patterns, especially in response to regulations, such as the ADEC burn restrictions, is a complex issue that requires appropriately acknowledging the influence of meteorology, the physicochemical nature of the emissions, and change in emissions at the source. Based on the observations in this study, it cannot be conclusively inferred that either hardwood- or softwood-based solid fuels are more popularly consumed wood types in Fairbanks, but they are certainly among the largest contributors to sub-micron OA

emissions. These findings are critical to addressing air pollution in Fairbanks, which has been a persistent issue for a long time, by guiding policies and citizen action.

**Data availability**

Supporting text, figures, and tables are available in the Supplementary Material.



## Author contributions

The manuscript was written with the contributions of all authors. BT-R and BD set up, ran, and maintained the instrumentation during the campaign in Fairbanks. SA, NB, and ED aided during the campaign. MC-S collected and contributed meteorological and trace gas data. BA, RJW, KD, and AM provided data on ion chromatography analysis of offline filter samples. BT-R and AI processed and analysed the data with help from BC. WS and KS coordinated the ALPACA and CASPA projects. KL, BD, BB, SB, JF, JM, and JS contributed to funding acquisition 675 for the CASPA project. BD supervised the project reported here.

## Competing interests

The authors declare that they have no conflict of interest.

## Acknowledgements

We thank the ALPACA team of researchers and all others involved in designing the project and providing the 680 necessary logistical support to carry it out. We thank Dr Anna Tobler (Datalystica Ltd, Switzerland) for providing technical support for SoFi analysis. This work was funded by the CASPA (Climate-relevant Aerosol Sources and Processes in the Arctic) project of the Agence Nationale de la Recherche (grant no. ANR-21-CE01-0017) and the IPEV (French Polar Institute Paul-Émile Victor). KD and RJW were supported by the National Science Foundation's (NSF) Atmospheric Geoscience Program (grant no. AGS-2029730) and the NSF Navigating the New 685 Arctic Program (grant no. NNA-1927778). SRA acknowledges support from the UK Natural Environment Research Council (grant ref: NE/W00609X/1). We thank the MASSALYA instrumental platform (Aix-Marseille Université, Laboratory of Chemistry and Environment, lce.univ-amu.fr) for the measurements used in this publication. AI is grateful to Subuktageen Qitta, Nastaran Mahmud, Laal Boo'on-Wali, and Minuit Mahmud for the many useful discussions that helped compile this report.

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





**Figure 1** Overview of meteorological parameters and aerosol properties. The shaded areas show the periods, when Stage 1 (red) and 2 (black) advisories ("burn bans") from the Alaska Department of Environmental Conservation, were in place in Fairbanks. (**A**) Ambient temperature at 3 and 23 m and difference of temperature between the two heights; (**B**) wind speed and direction with the daily sunlight in terms of the $NO_2$ photolysis rate coefficient ($J_{NO2}$); (**C–D**) absolute and fractional compositions of composition of non-refractory fine particulate matter (NR-PM$_1$) from the AMS; and (**E**) size distribution of PM$_1$ from the SMPS and its comparison with PM$_{2.5}$.



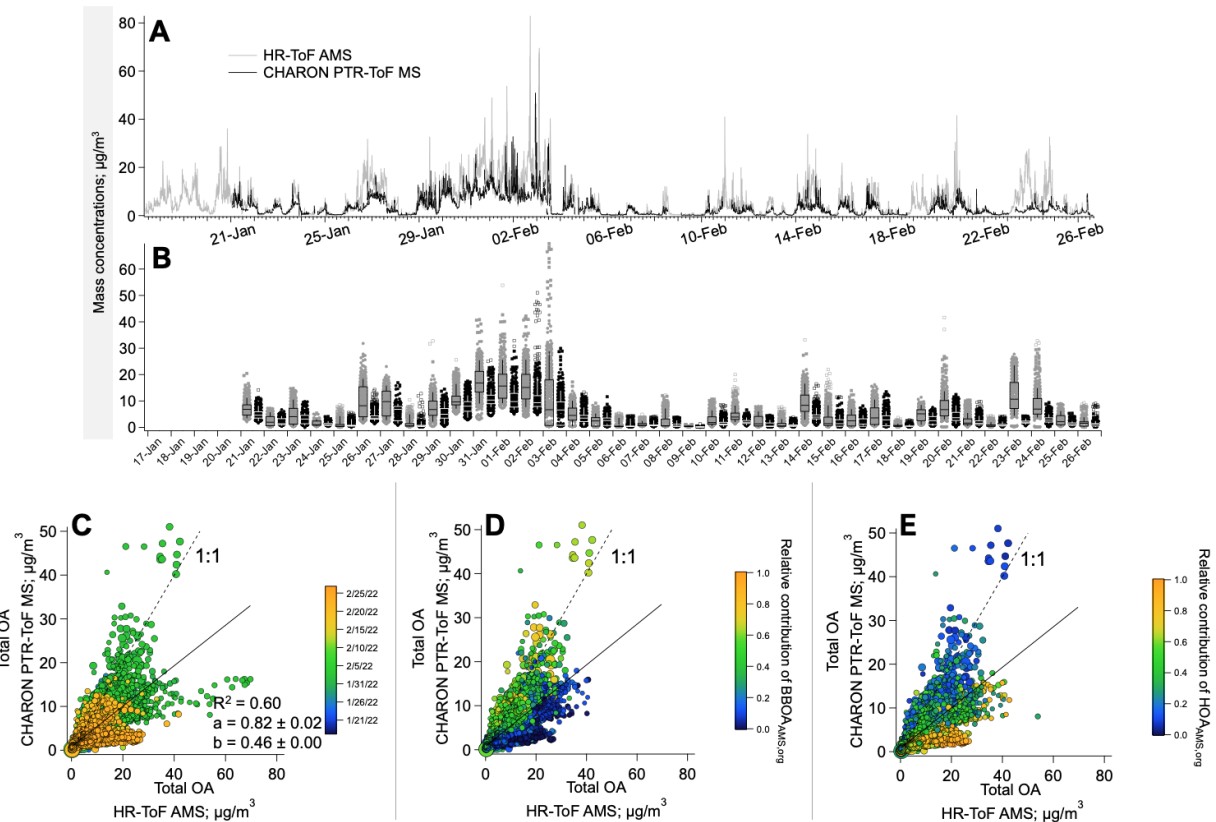

**Figure 2** Comparison of total OA measured with the PTR$_{CHARON}$ and the AMS. (**A**) Absolute concentrations of OA measured with the AMS and OA$_{corr}$ (fragmentation-corrected OA) from PTR$_{CHARON}$; (**B**) Daily average concentrations of OA; (**C**) Scatter plot of total OA measured with the AMS and the PTR$_{CHARON}$. Data points are coloured by the dates and the legend is written as MM/DD/YY. Data points are sized by the geometric mean mass of the dM/dlogDp from SMPS (50–500 nm). The dashed line denotes the 1:1 relationship. Coefficients, *a* and *b*, denote the slope and the intercept for the linear regression ($p \leq 0.05$; solid line) and are written with ± one standard deviation; (**D–E**) The scatter plot in panel (**C**) is redrawn with different colours, i.e., the relative contribution of biomass burning OA and hydrocarbon-like OA factors diagnosed in AMS analysis.





**Figure 3** Overview of the positive matrix factorisation output for NR-PM$_1$ measurements with the AMS (called AMS$_{org+inorg}$ in-text). The normalised mass spectra, time series, and diurnal patterns are shown for six factors diagnosed. Mass spectra are coloured by the elemental composition of the fragments. Mass concentrations were normalised to the sum of the concentrations of all ions. Time series are overlaid with those of the corresponding factor (if available) in AMS$_{org}$ and PTR$_{CHARON}$ analysis or an external tracer. Correlation coefficients ($R^2$; $p \leq 0.05$) are also provided and slopes can be found in **Table S5** or **Table**



**Figure 4** Normalised mass spectra of factors from the positive matrix factorisation of PTR$_{\mathrm{CHARON}}$ measurements. Mass concentrations are normalised to the sum of concentrations of all ions. Peaks are coloured by the molecular group (CHO, CHNO, CHOS, CH, CHN) of the formula assigned. Unassigned species are shown in black. Further information, such as tentative identities and formula errors, can be found in **Supplementary Dataset 1**.







**Figure 5** Diurnal profiles and complete time series of factors from the positive matrix factorisation of PTR$_{CHARON}$ measurements. In the second column, time series are overlaid on those of the corresponding factor in AMS$_{org}$ and an external tracer or marker ion. Scatter plots depict the temporal correlations ($p \leq 0.05$) between OA mass concentrations measured with the AMS and PTR$_{CHARON}$. Details on the correlations with the external tracers can be found in **Table 1**.



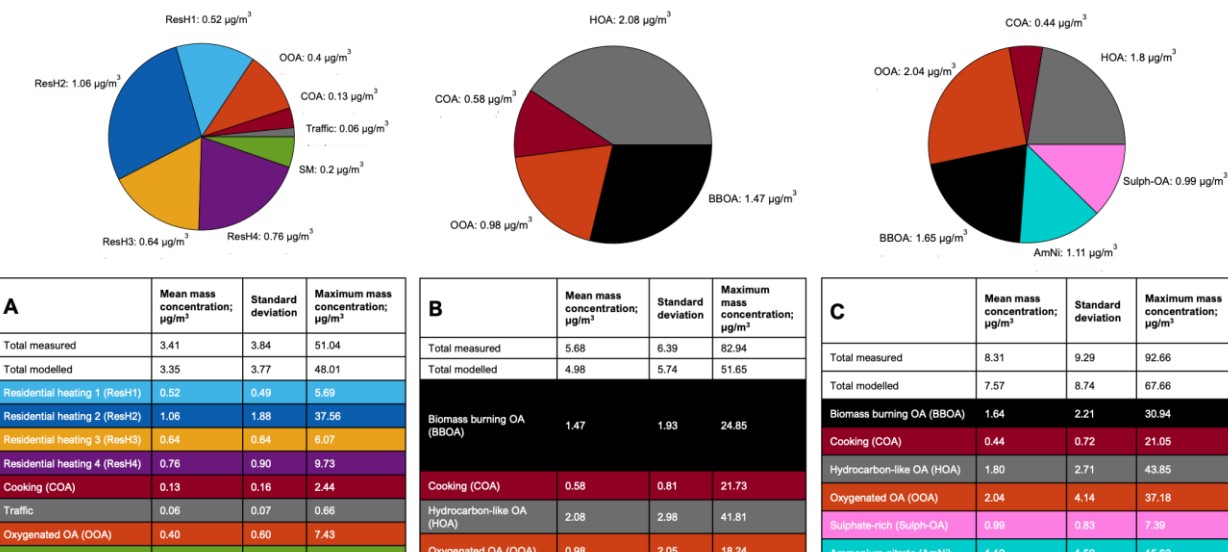

**Figure 6** Campaign-averages of mass concentrations apportioned to each factor in (**A**) PTR$_{CHARON}$, (**B**) AMS$_{org}$, and (**C**) AMS$_{org+inorg}$ analyses. Slices of pies are equivalent to the average absolute concentrations. A complete time series of fractional contributions can be found in **Figure S14**.





**Figure 7** Scatter plots showing the correlation ($R^2$; $p \leq 0.05$) between inorganic species measured with the AMS and offline ion chromatography of chemical species in $PM_{0.7}$ collected on filters. Comparison of (**A**) total mass concentrations of sulphur and nitrogen-containing species; (**B**) $OOA_{AMS,org+inorg}$ factor with different species from IC analysis; and (**C**) Sulph-OA factor with different species from IC analysis.



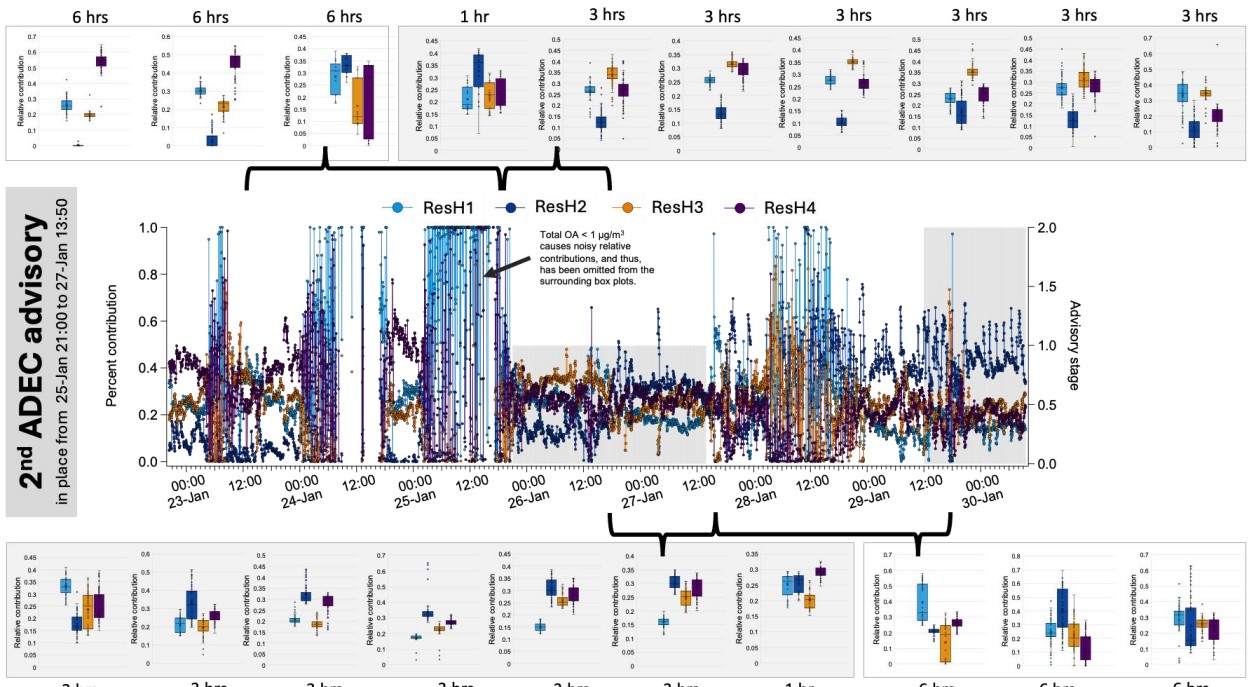

**Figure 8** Variation in the relative contributions of residential heating factors to total biomass-burning OA concentrations. For simplicity, only the 2nd ADEC advisory implemented during the campaign is shown. Contributions are also shown for approximately 2 days before and after the advisory for comparison, along with their 6-hour averages as box plots (white panels), when suitable data was available (e.g., periods with noisy data were omitted and the adjacent period is shown instead). For better visualisation of variation in contributions, when the advisory was in place, 3-hour averages are shown (grey panels). To account for a lag in the appearance of variations in emission sources, 1-hour averages are shown for the beginning and end of the advisory event.





**Table 1** Linear regression ($R^2$; $p \leq 0.05$) between the time series of factors derived from (**A**) PTR$_{CHARON}$, (**B**) AMS$_{org}$, and (**C**) AMS$_{org+inorg}$ measurements with external tracers and chemical species (S and N-containing species and PAHs) measured with the AMS.

**A**

| | Traffic | COA | OOA | ResH1 | ResH2 | ResH3 | ResH4 | SM |
|---|---|---|---|---|---|---|---|---|
| Amb. Temp. | 0.01 | 0.02 | 0.22 | 0.14 | 0.27 | 0.27 | 0.20 | 0.16 |
| Black carbon | 0.58 | 0.27 | 0.22 | 0.37 | 0.16 | 0.27 | 0.22 | 0.04 |
| **Trace gases** | | | | | | | | |
| $NO_2$ | 0.46 | 0.19 | 0.26 | 0.37 | 0.15 | 0.27 | 0.16 | 0.01 |
| NO | 0.65 | 0.24 | 0.22 | 0.32 | 0.10 | 0.16 | 0.13 | 0.06 |
| $NO_x$ | 0.66 | 0.25 | 0.25 | 0.36 | 0.12 | 0.20 | 0.15 | 0.05 |
| $CO_2$ | 0.67 | 0.38 | 0.31 | 0.51 | 0.24 | 0.39 | 0.30 | 0.02 |
| CO | 0.61 | 0.18 | 0.08 | 0.14 | 0.02 | 0.04 | 0.03 | 0.08 |
| $SO_2$ | 0.27 | 0.20 | 0.19 | 0.46 | 0.34 | 0.61 | 0.47 | 0.01 |
| $O_3$ | 0.34 | 0.19 | 0.13 | 0.39 | 0.12 | 0.31 | 0.20 | 0.00 |
| **Chemical species measured with the HR-ToF AMS** | | | | | | | | |
| Sulphur | 0.43 | 0.22 | 0.71 | 0.35 | 0.22 | 0.23 | 0.13 | 0.04 |
| $NO_3$ | 0.31 | 0.16 | 0.25 | 0.17 | 0.02 | 0.04 | 0.01 | 0.02 |
| $NH_4$ | 0.43 | 0.20 | 0.64 | 0.30 | 0.15 | 0.14 | 0.06 | 0.05 |
| Cl | 0.10 | 0.05 | 0.12 | 0.06 | 0.01 | 0.03 | 0.01 | 0.01 |
| UnSub PAH | 0.30 | 0.25 | 0.34 | 0.50 | 0.59 | 0.55 | 0.58 | 0.01 |
| M-PAH | 0.33 | 0.27 | 0.33 | 0.52 | 0.60 | 0.53 | 0.60 | 0.01 |
| O-PAH | 0.27 | 0.22 | 0.36 | 0.56 | 0.70 | 0.61 | 0.64 | 0.01 |
| N-PAH | 0.28 | 0.23 | 0.26 | 0.54 | 0.62 | 0.61 | 0.68 | 0.01 |
| A-PAH | 0.28 | 0.24 | 0.19 | 0.48 | 0.50 | 0.55 | 0.61 | 0.04 |

**B**

| | HOA | COA | OOA | BBOA |
|---|---|---|---|---|
| Amb. Temp. | 0.02 | 0.02 | 0.19 | 0.22 |
| Black carbon | 0.49 | 0.27 | 0.29 | 0.25 |
| **Trace gases** | | | | |
| $NO_2$ | 0.42 | 0.25 | 0.25 | 0.25 |
| NO | 0.61 | 0.26 | 0.33 | 0.16 |
| $NO_x$ | 0.62 | 0.28 | 0.34 | 0.20 |
| $CO_2$ | 0.49 | 0.30 | 0.41 | 0.35 |
| CO | 0.38 | 0.19 | 0.19 | 0.06 |
| $SO_2$ | 0.18 | 0.14 | 0.25 | 0.44 |
| $O_3$ | 0.26 | 0.20 | 0.12 | 0.27 |
| **Chemical species measured with the HR-ToF AMS** | | | | |
| Sulphur | 0.37 | 0.27 | 0.89 | 0.19 |
| $NO_3$ | 0.49 | 0.27 | 0.23 | 0.06 |
| $NH_4$ | 0.48 | 0.29 | 0.79 | 0.13 |
| Cl | 0.12 | 0.06 | 0.13 | 0.03 |
| UnSub PAH | 0.31 | 0.26 | 0.39 | 0.71 |
| M-PAH | 0.36 | 0.30 | 0.39 | 0.76 |
| O-PAH | 0.23 | 0.23 | 0.43 | 0.79 |
| N-PAH | 0.24 | 0.22 | 0.33 | 0.78 |
| A-PAH | 0.23 | 0.20 | 0.26 | 0.69 |

| | |
|---|---|
| Very strong | ≥0.75 |
| Strong | ≥0.5 and <0.75 |
| Moderate | ≥0.3 and <0.5 |
| Weak | ≥0.1 and <0.3 |
| None | <0.1 |

**C**

| | HOA | COA | OOA | BBOA | AmNi | Sulph-OA |
|---|---|---|---|---|---|---|
| Amb. Temp. | 0.01 | 0.03 | 0.19 | 0.26 | 0.00 | 0.24 |
| Black carbon | 0.43 | 0.21 | 0.32 | 0.30 | 0.30 | 0.30 |
| **Trace gases** | | | | | | |
| $NO_2$ | 0.37 | 0.18 | 0.27 | 0.28 | 0.31 | 0.40 |
| NO | 0.55 | 0.19 | 0.36 | 0.22 | 0.35 | 0.32 |
| $NO_x$ | 0.56 | 0.21 | 0.37 | 0.26 | 0.38 | 0.37 |
| $CO_2$ | 0.41 | 0.24 | 0.47 | 0.41 | 0.28 | 0.48 |
| CO | 0.35 | 0.17 | 0.21 | 0.08 | 0.25 | 0.11 |
| $SO_2$ | 0.14 | 0.11 | 0.27 | 0.45 | 0.07 | 0.61 |
| $O_3$ | 0.23 | 0.14 | 0.12 | 0.25 | 0.23 | 0.34 |
| **Chemical species measured with the HR-ToF AMS** | | | | | | |
| Sulphur | 0.26 | 0.24 | 0.95 | 0.34 | 0.23 | 0.48 |
| $NO_3$ | 0.38 | 0.18 | 0.24 | 0.09 | 0.98 | 0.12 |
| $NH_4$ | 0.34 | 0.25 | 0.86 | 0.25 | 0.44 | 0.33 |
| Cl | 0.09 | 0.05 | 0.15 | 0.04 | 0.16 | 0.04 |
| UnSub PAH | 0.26 | 0.25 | 0.40 | 0.77 | 0.15 | 0.31 |
| M-PAH | 0.31 | 0.28 | 0.40 | 0.82 | 0.17 | 0.32 |
| O-PAH | 0.18 | 0.23 | 0.41 | 0.87 | 0.11 | 0.33 |
| N-PAH | 0.20 | 0.20 | 0.33 | 0.82 | 0.11 | 0.30 |
| A-PAH | 0.20 | 0.18 | 0.26 | 0.70 | 0.11 | 0.25 |