# Peer review of "of wintertime sub-micron particle pollution in Fairbanks,"

_EGUsphere, 2024_

## Author Comment (AC1)

[revised manuscript text omitted]
 identified and quantified various local sources using source apportionment approach and could determine the their chemical composition, size distribution, and concentrations of atmospheric sub-micron non-refractory particulate matter (NR-PM$_1$) emitted from them and quantify their sources in the urban centre of Fairbanks. As part of the Alaskan Layered Pollution and Chemical Analysis (ALPACA) campaign, we deployed a Chemical Analysis of Aerosol Online (CHARON) inlet coupled with a proton transfer reaction - time of flight mass spectrometer (PTR-ToF MS) and an Aerodyne high-resolution aerosol mass spectrometer (AMS) to measure organic aerosol (OA) and NR-PM$_1$, respectively. We deployed Positive Matrix Factorisation (PMF) analysis for source identification of the NR-PM$_1$. The PTR$_{CHARON}$ factorisation identified four residential heating sources, with oil combustion accounting for 16 ± 9% and wood combustion contributing to 47 ± 20% of the OA. The analysis could further differentiate between hardwood and softwood combustion. In contrast, The AMS analysis identified three primary factors: a single biomass burning, a hydrocarbon-like and cooking factors which together accounted for 28, 38 and 11 % of the total OA, respectively. Additionally, a combined organic and inorganic PMF analysis revealed two further factors: one enriched in nitrates and another rich in sulphates of organic and inorganic origin. The PTR$_{CHARON}$ factorization could identify four primary sources from residential heating - one from oil combustion and three wood combustion, categorised as types (low temperature, softwood, and hardwood). Collectively, all residential heating factors accounted for 79% of the total OA. cCooking and road transport were also recognised as primary contributors to overall emission profile provided by PTR$_{CHARON}$. All PMF could apportion a single secondary organic fraction. These results evidence the complementarity of the two instruments and their ability in describing the complex chemical composition of PM$_1$ and the related sources. These results demonstrate that This work further demonstrates the capability of PTR$_{CHARON}$ to provide both generates qualitative and quantitative information offering with enhanceda comprehensive understanding of the organic aerosol sources. resolution of organic aerosol sources. Such 
[revised manuscript text omitted]
 traffictransport$_{CHARON}$ factor (**Figure 6**). We speculate that the These discrepancies shortcomings seen in OA mass measuredments by the from $PTR_{CHARON}$ relative compared to the $AMS_{org}$ wereare largely instrumental, particularly partly duedue to such as the poor transmission low sensitivities of the $PTR_{CHARON}$ for small particles (<100 nm) by the CHARON inlet and the limited sensitivity towards hydrocarbons by PTR, but other possible biases can be due to heating oil OA signal interfering with the $HOA_{AMS}$, as discussed in **S8**. Previous studies using the $PTR_{CHARON}$ in Innsbruck, Austria, successfully observed a traffic factor, but did not detect no cooking emissions despite sampling operating at in an urban locality area (MüLler et al., 2017). A vVarietyous of environmental and user biases could also may be involved, such as the contribution of non-vehicular sources to the $HOA_{AMS}$ factors and the choiceuse of suboptimal conversion coefficients (e.g., RIE) in the AMS analyses (see **Sections S8** and **S9** for details). These are important considerations are critical for ambient air analysis in employing the with $PTR_{CHARON}$ , as obtaining for ambient air analyses because a full picture of the sources involved, especially in urban regions influenced by

**3.2.4. Cooking organic aerosol**

[revised manuscript text omitted]

indicating a significant content of sulphur in sulphurcontaining compounds (**Figure S15**). The AMS does not quantitatively distinguish among the different sulphur-containing species, but such as hydroxymethane sulphonate (HMS; $CH_2(OH)SO^{-3}$), $SO_3^{2-}$ (sulphite), $HSO_3^-$ (bisulphite), and $SO_4^{2-}$ (sulphate), or between organic and inorganic sulphur, but we have tried to use following guidelines from previous works (Chen et al., 2019; Schueneman et al., 2021) we could explored tthe ratio of sulphur fragments to investigate speculate on the presence of different species forms of sulphur present such as hydroxymethane sulphonate (HMS; $CH_2(OH)SO^{-3}$), $HSO_4^-$ (bisulphite), $SO_4^{2-}$ (sulphate) and (H2SO4) sulphuric acid. In light of previous studies on sulphur source apportionment with the AMS and source differentiation based on fragmentation patterns, (Chen et al., 2019; Schueneman et al., 2021).An organosulphate content of ~ **0.8 ± 1.3 μg/m³** (**~20 ± 16%**) was then derived using the ratios of $SO^+$ and $SO_2^+$ ions against $SO_3^+$, $HSO_3^+$, and $H_2SO_4^+$ ions as detailed by Song et al., (2019). This value is in good agreement with previous reports from the same field campaign (Campbell et al., 2022; Robinson et al., 2024). Additionally, to mimic potential matrix effects of wood burning OA on sulphate fragmentation patterns, we performed calibrations on the AMS spectra from a solution of with pure $(NH_4)_2SO_4$ mixed with various amounts of levoglucosan (i.e., 0–80% in mass). This mixture was used to assess the matrix effect that can potentially impact sulphur fragmentation patterns in an environment affected by wood smoke as previously demonstrated with other organic matrices. We were compared to ambient data and PMF factors as shown compared the fractions of $HSO_3^+$ to $H_2SO_4^+$ fragments normalised to the fractions $H_2SO_4^+$ and $HSO_3^+$ for pure $(NH_4)_2SO_4$ as suggested by Chen et al., (2019). Results are shown in **Figure S16A**. Results Among the sulphate-rich factors the are shown in **Figure S16A** where the $OOA_{AMS,org+inorg}$ factor exhibited much lower $HSO_3^+$ to $H_2SO_4^+$ intensities suggesting which is indicative of an influence of an higher fraction of organosulphate -compounds, as it is also evidenced ininfluence **Figure S16D–E** by the strong correlation between derived organo-sulpahur fraction and sulphate-ions in the $OOA_{AMS,org+inorg}$ factor ($R^2$ = 0.85, slope = 0.57). (**Figure S16D–E**). The average total concentration of sulphursulphate-related fragments in $OOA_{AMS,org+inorg}$ was **0.9 ± 1.8 μg/m³**, on average, and which accounted for **26 ± 23%** of the total sulphur sulphate measured with the AMS. This finding is consistent , which agrees with the theoretical estimations of organosulphur content (Song et al., 2019).

Further information on chemical composition was gathered by comparing this factorAMS results with IC measurements from $PM_{0.7}$ filters sampleds analysed as part of another ALPACA study (Dingilian et al., 2024). Despite the good correlations between the two datasets, shown in **Figure 7A,** the AMS results underestimated - Although there is a notable underestimation Both methods (IC and AMS) correlated well, despite a negative bias against in the AMS analysis that underestimated the sums of approximately by 30%, 26%, and 35% lower sulphursulphate, ammonium, and nitrate-related fragments, respectively (see **Section 2.2.2** for fragments included). the correlation between the two

~~methods remains strong by ~ 31, 26%, and 35% compared to the IC analyses (**Figure 7A**).very strong(**Figures 7B and S16F–I**)with a relatively, but still strong,In additionAdditionally, thethisverytotal(95**S16D–E**under favourable meteorological conditions and aerosol compositionhave beenobserved thewith 
[revised manuscript text omitted]

**Response to **Reviewer 1**

**This manuscript reports and discusses the source apportionment of organic and inorganic aerosols in Fairbanks, Alaska during winter, as measured by HR-ToF-AMS and CHARON-PTR-ToF-MS. Fairbanks is of particular interest as during winter, it frequently has poor air quality due to enhanced biomass burning emissions (residential heating) combined with poor air dispersion caused by temperature inversions. The focus of the paper is on comparison of the source apportionment results from the AMS and CHARON-PTR data, emphasizing their complementary nature. While AMS data have been extensively used for source apportionment, that is not the case for CHARON-PTR or even PTR-MS. To my knowledge, there is only one (very recent) source apportionment study that uses both CHARON-PTR and AMS data. The paper by Ijaz et al provides a very detailed source apportionment study and it will serve as a useful resource for future source apportionment studies using both CHARON-PTR and AMS data. It also demonstrates the ability of CHARON-PTR, due to its molecular level characterisation of OA, to distinguish additional residential heating sources. This is something that would not be possible to identify based only on the AMS data. Based on the arguments outlined above I recommend this paper for publication in the ACP.**

→ Thank you for your thoughtful review and valuable feedback! We appreciate your acknowledgment of the complementary roles of AMS and CHARON PTR-ToF MS datasets in source apportionment. We look forward to incorporating your suggestions to enhance our manuscript and are grateful for your recommendation for publication in *ACP*. For your information, due to the request from reviewer 2, we did revise and reformulate some parts of the article (abstract, results, and conclusion).

**Below are my specific comments (most of which I would describe as minor).**

**Line 120: I suggest "molecular level information" instead of "good molecular resolution"**

→ We have revised the manuscript to incorporate "molecular level information" instead of "good molecular resolution" to enhance clarity.

**Line 134 Please use sulfate instead of sulfur.**

→ The revised manuscript reflects the change from sulphur to sulphate, where appropriate, as requested (**multiple lines along the document**).

**Line 174: what was *m/z* range?**

→ The *m/z* range applied was *m/z* 50–425. This has now been specified as follows:

***"Among the resolved 1118 ions spanning the range of m/z 50–425, only 336 were retained above the S/N, and 318 ions could be given a molecular formula based on the criteria described in Section S2." (section 2.2.1 now line 193)***

**Line 188: no need for “PeTeR version 6.0…” as it has already been mentioned**

→ We have removed the redundant mention in the revised manuscript.

**Line 195: As AMS’ come with either capture or standard vaporiser, it is worth mentioning which one you had.**

→ The HR-ToF-AMS used in this study is equipped with a standard vapouriser. We have modified the sentence as follows: ***“Briefly, ambient particles are sampled through a critical orifice, focused into a narrow beam by an aerodynamic lens, accelerated toward a standard vapouriser heated at 600°C, and then ionised by electron impact (70 eV at $10^{-7}$ torr).”***

**Line 199: ..nitrate, ammonium, sulfate, chloride.**

→ The revised manuscript reflects the change from sulphur to sulphate throughout the document.

**Line 208-9: nitrate, sulfate, chloride (please use singular for these ions).**

→ These species have now been written in singular forms (**section 2.2.2**).

**Line 2016: why max m/z was only 120; why were PAHs excluded?**

→ Our initial analysis incorporated all PAHs in the AMS analysis, and the resulting factorisation was the same as what we presented in the paper, achieving the primary aim. Another possible concern regarding the removal of PAHs is the potential underestimation of the mass concentration attributed to the factors. However, ~98% of AMS signal was present in fragments with *m/z* <120. Still, this limitation was acknowledged in the original manuscript as follows: ***“This exclusion is expected to cause underestimation (by <2%) of the mass of some factors, particularly HOA (hydrocarbon-like organic aerosol) and BBOA (biomass-burning organic aerosol)”.***

→ First we extracted PAHs from AMS measurements to be utilized as external tracers for $PTR_{CHARON}$ analysis, where they are not quantified as clearly, and were necessary for a better understanding of the factors. Specifically, we used them to differentiate factors like BBOA and HOA from other factors. Since the external tracers used for measuring external correlations of factors from $PTR_{CHARON}$ and AMS analysis were kept consistent for the sake of data comparability, it was redundant to include PAHs in the PMF analysis of the AMS measurements.

Furthermore, many PMF analyses restrict the high-resolution organic mass spectra to *m/z* 120 (e.g., Aiken et al, https://doi.org/10.5194/acp-9-6633-2009; Young et al, https://doi.org/10.5194/acp-16-5427-2016, Duan et al, https://doi.org/10.5194/acp-22-10139-2022; Qi et al., https://doi.org/10.1016/j.scitotenv.2021.151800, among many others).

This approach is warranted because most of the signal from OA and inorganic species is produced within this mass range, largely due to extensive fragmentation resulting from electronic impact ionization of larger molecules. So finally, we extracted PAHs from AMS measurements to utilise them as external tracers (Figure 5). Specifically, we used them to differentiate factors like BBOA and HOA from other factors.

**Line 220: not needed to mention that empty rows and columns were removed (same for line 188).**

→ We agree that the phrase was redundant. It has now been removed from the revised manuscript.

**Line 271: have you tried to look at the correlation between aerosol loadings and WS for the periods with strong temperature inversion? This would be a nice visual confirmation of your statement.**

→ Yes, Figure 1 depicts the wind speed, as well as wind direction and temperature inversion, and their correlation with aerosol loading and particle number concentration throughout the campaign. This figure and accompanying text are emphasized in section 3.1. Figure 1 description:

***"Figure 1 depicts meteorological conditions, chemical composition and particle size distribution of NR-$PM_1$ observed from January 20 to February 26, 2022. Intense aerosol loads coincided with poor atmospheric dispersion, associated with low wind speeds (<2 m/s), low temperature (below - 10°C) associated with strong surface temperature inversions, with temperature differences between 23 and 3 m above sea level ranging from 3°C to 10°C)."***

**Line 272: low instead of slow**

→ Thank you for pointing out the error. We have corrected (see sentence above).

**Line 276: daily average instead of campaign-averaged?**

→ The term, campaign-average, is used because it means something else than daily average. We also have mentioned daily averages specifically for the pollution episodes. It is important to mention both to clearly illustrate how the pollution episodes differed from the overall campaign average.

**Line 284: sulfate instead of sulfur. I am assuming the authors chose to use sulfur as species other than $SO_4^{2-}$ can result in fragments that are used to calculate the overall $SO_4^{2-}$ mass loadings (e.g. organosulfates). The choice of naming should be either explained in the Exp section, or use sulfate and discuss the contribution of organosulfates to $SO_4^{-2}$ later in the text (page 15). Note that Fig 1 has SO4, not S.**

→ We agree with the reviewer. Our reasoning behind using "sulphur," rather than "sulphate" was indeed the possibility of the contribution from non-$SO_4^{2-}$ species. However, considering the traditional use of "sulphate" in the AMS community, we have corrected in the revised manuscript (multiple lines).

**Line 318: fragmentation increases dramatically when?**

→ Thank you for pointing out this phrase. This was an improperly phrased sentence. We have now rephrased it to "***Tests conducted in our laboratory with five $C_{16}$–$C_{26}$ alkanes as markers of vehicle emissions revealed that they undergo extensive fragmentation, resulting in 2–4 times underestimation of their actual concentrations. In line with this finding, the ineffective***

***ionisation of saturated alkanes by PTR (Ellis and Mayhew, 2014) and their tendency to undergo dissociative ionisation (Gueneron et al., 2015) has also been reported previously. "***

**Line 359: Considering that ResH1_4 "closely co-varied in time", how strongly do they correlate with each other? Could they all, or some of them be a part of the same factor (i.e. be a result of factor splitting)?**

→ We understand the reviewer's concerns about the complexity of the 8-factor solution and the potential for co-varying factors. However, we would like to assert that we have thoroughly evaluated the PMF results and molecular compositions to separate residential heating sources accurately. Table S2 presents a summary of PMF results from 3 to 10 factors. In the $PTR_{CHARON}$ a splitting is observed for more than 8 factors, while below 8 factors, we could see mixing of ResH4 (hardwood) and ResH3 (heating oil), which are different sources based on their molecular compositions and correlation with $SO_2$.

Also the time correlation of the 4 residential factors is not as good as we could suppose for split factors ($R^2$ from 0.35 to 0.55) as can be observed in the table below.

| A | Traffic | COA | OOA | RH1 | RH2 | RH3 | RH4 | SM |
|---|---|---|---|---|---|---|---|---|
| Traffic | 1.00 | | | | | | | |
| COA | **0.29**
a=0.04 ± 0.00
b=1.20 ± 0.01 | 1.00 | | | | | | |
| OOA | **0.17**
a=0.17 ± 0.00
b=3.42 ± 0.06 | **0.11**
a=0.25 ± 0.00
b=1.24 ± 0.03 | 1.00 | | | | | |
| RH1 | **0.27**
a=0.29 ± 0.00
b=3.55 ± 0.04 | **0.19**
a=0.38 ± 0.00
b=1.33 ± 0.02 | **0.18**
a=0.40 ± 0.00
b=0.34 ± 0.00 | 1.00 | | | | |
| RH2 | **0.07**
a=0.58 ± 0.02
b=6.87 ± 0.20 | **0.08**
a=0.67 ± 0.02
b=3.22 ± 0.09 | **0.15**
a=0.57 ± 0.02
b=1.16 ± 0.02 | **0.36**
a=-0.23 ± 0.02
b=2.30 ± 0.02 | 1.00 | | | |
| RH3 | **0.10**
a=0.44 ± 0.01
b=2.81 ± 0.06 | **0.12**
a=0.48 ± 0.01
b=1.28 ± 0.03 | **0.12**
a=0.49 ± 0.01
b=0.36 ± 0.01 | **0.52**
a=0.1 ± 0.00
b=0.94 ± 0.01 | **0.41**
a=0.43 ± 0.00
b=0.21 ± 0.00 | 1.00 | | |
| RH4 | **0.08**
a=0.50 ± 0.01
b=3.48 ± 0.10 | **0.11**
a=0.52 ± 0.01
b=1.74 ± 0.04 | **0.06**
a=0.61 ± 0.01
b=0.34 ± 0.01 | **0.35**
a=0.09 ± 0.01
b=1.08 ± 0.01 | **0.50**
a=0.39 ± 0.01
b=0.32 ± 0.00 | **0.55**
a=0.03 ± 0.01
b=1.05 ± 0.01 | 1.00 | |
| SM | **0.06**
a=0.16 ± 0.00
b=0.47 ± 0.01 | **0.05**
a=0.29 ± 0.00
b=3.55 ± 0.04 | **0.02**
a=0.18 ± 0.00
b=0.03 ± 0.00 | **0.00**
a=0.19 ± 0.00
b=0.00 ± 0.00 | **0.01**
a=0.18 ± 0.00
b=0.01 ± 0.00 | **0.00**
a=0.19 ± 0.00
b=-0.00 ± 0.00 | **0.01**
a=0.66 ± 0.01
b=0.51 ± 0.06 | **1.00** |

Correlations with AMS BBOA factor are shown in **Table S4 ($R^2$ = 0.5–0.7).** These temporal correlations are expected, as different types of residential heating are likely to be synchronised in time due to similar public needs for heating during the wintertime, despite potential differences in fuel usage.

Furthermore, when combining OA PMF-charon with size distribution data, we see that the factors are associated with particles having distinct size distributions (notably ReshH3 is associated with smaller particles than the other wood combustion factors, **see Figure S12 plot A**). This supports our choice of an 8-factor solution as the most stable representation of the data.

**Line 369: what is in the same order? I assume peak intensity?**

→ Yes, correct. The signal (or peak intensity!) of levoglucosan and its fragments were distributed in ResH1, ResH4, and ResH2 in the same order as the factors listed, i.e., most signal belonged to ResH1 and the least to ResH2. However, "in the same order" appears to be confusing and unnecessary in this instance, and thus, has been removed.

**Line 385: atmospheric processing. Could it be that ResH1 is related more to low-temperature combustion?**

→ Yes the reviewer is right, actually in a first version of the article we discussed it and then it was removed it. But we do agree the ResH1 factors presents considerable high content of oxygenated compounds and have a relatively high value of O/C ratio (0.42), among the most abundant molecules we observe furans, furfurals etc. We therefore modified the paragraph as follows:

**"ResH1 includes low temperature combustion markers: this factor is small as it contributes to only an average of 0.5 ± 0.5 μg/m³ of the total $OA_{CHARON}$ concentration, but it contains the highest fraction of levoglucosan (~30%). Approximately 65% of the total signal of ResH1 is due to compounds with six or fewer carbon atoms, compared to heavier species present in the other factors (Figure S13) and the most abundant species are at m/z 69.03 ($C_4H_4O$; furan) (Palm et al., 2020; Jiang et al., 2019), m/z 87.04 ($C_4H_6O_2$; oxobutanal) (Brégonzio-Rozier et al., 2015), m/z 97.03 ($C_5H_4O_2$; furfural), m/z 109.0286 ($C_6H_4O_2$; benzoquinone) (Stefenelli et al., 2019b) and m/z 115.04 ($C_5H_6O_3$; methyldihydrofuran) (Koss et al., 2018). Consistent with these molecular formulae, the concentration-weighted average O/C of ResH1 was relatively higher (i.e., 0.42) compared to other residential heating factors (O/C = 0.2–0.3). The most abundant species observed in ResH1 can be related to depolymerisation reactions occurring during low temperature and early stages of the combustion process(Collard and Blin, 2014; Sekimoto et al., 2018)."**

**Line 483: CHARON can see species that can evaporate at 150ºC (I am assuming this is the thermodenuder temp.; please mention operating TD temperature in the Experimental section); there are also species that will not be efficiently ionised by proton transfer. This should also be mentioned as a potential reason for the observed discrepancy.**

→ We agree with your perspective and have clarified this aspect in the Methodology section by specifying the thermodenuder operating temperature as follows: "***The thermodesorber in the CHARON inlet was operated at 150°C and 8 mbar; this combination of moderate temperature with low pressure expands the range of detection to include ELVOCs as well*** **(Piel et al., 2021)."**

To address the potential reasons for discrepancies between AMS and $PTR_{CHARON}$, we have revised the text as follows: **"*Part of the quantitative difference between the two instruments can also be explained by the PTR limitation in ionisation and the ionisation-induced fragmentation of analyte ions. Tests conducted in our laboratory with five C16–C26 alkanes as markers of vehicle emissions revealed that they undergo extensive fragmentation, resulting in 2–4 times underestimation of their actual concentrations. In line with this, the ineffective ionisation of saturated alkanes by PTR (Ellis and Mayhew, 2014) and their tendency to undergo dissociative PTR ionisation (Gueneron et al., 2015) has also been reported previously.*"**

**how do overall mass loadings of AMS Org and Charon Org compare? Are there any other studies reporting concurrent measurements by CHARON and AMS? How did their mass loadings compare? And doing source apportionment? If yes, you did their factor loadings compare?**

→ The difference in OA mass loadings measured by the AMS and $PTR_{CHARON}$ have been thoroughly discussed in our paper, particularly in Section 3.1 and shown in Figures 2 and S9. Briefly, the OA mass loading from $PTR_{CHARON}$ was ~85% of the OA mass measured by the AMS. Although both instruments displayed a strong temporal agreement ($R^2 = 0.60$) as depicted in Figures 2A–B, the measurements were skewed either toward $AMS_{org}$ or the $PTR_{CHARON}$ during different periods of the campaign. These different periods were characterised by varying contributions from two primary sources: either HOA or BBOA. When BBOA contributed significantly (and thus, larger particles were present), $PTR_{CHARON}$ performed slightly better or comparably to the AMS; however, its performance declined during periods dominated by HOA emissions due to its inability to detect particles smaller than 100 nm and ineffective ionisation of long aliphatic chain molecules as explained to your previous comment.

These overall trends align with what we already know from previous studies, such as seminal papers by Muller et al, https://doi.org/10.1021/acs.analchem.7b02582), Leglise et al (https://doi.org/10.1021/acs.analchem.9b02949), and Song et al (https://doi.org/10.5194/acp-24-6699-2024). In these studies, AMS and $PTR_{CHARON}$ were compared, revealing that similar concentration variations were noted throughout the measurement intervals. In Muller et al. and Song et al, $PTR_{CHARON}$ detected ~88% and 62%, on average, of total OA mass measured by the AMS. Absolute concentrations were in agreement during certain times, while considerable differences were observed at other times, largely due to changes in emission characteristics and the relative abilities of the instruments to measure them.

As for the difference in mass loadings in source factors, Muller et al. and Song et al. conducted PMF on measurements from both instruments, but they did not provide direct numerical comparisons of concentrations between factors that were commonly identified in AMS and $PTR_{CHARON}$ analyses. Their focus was instead on gaining insights into source identities provided by the two instruments.

**Line 513: How well do OOA and BBOA correlate with each other?**

→ In the AMS measurements, there was a very weak correlation with $R^2 = 0.13$. Upon the inclusion of inorganics, the correlation becomes $R^2 = 0.26$. For $PTR_{CHARON}$, the $R^2$ values between OOA and the four residential heating factors ranged from 0.06–0.15. These values are presented in table S3 (time evolution correlation) and table S4 (spectral correlation).

**Line 524: I think it should be the other way around 0.4 and 0.2%.**

→ Thank you for catching this error! I rechecked our data on signal distribution across the factors and, to be exact, these were 0.48 and 0.29%. Other than the swapped sequence, these values also needed to be properly rounded off to 0.5 and 0.3%. The revision has been made.

**Line 591: remove "of"**

→ Thank you for catching the error. We have made the correction.

**Line 630: If the goal of this discussion and Fig 8 + S18-21 was to see which factor "responds" the most to ADEC advisories, then looking at relative contributions of ResH factors could be misleading as a decrease in absolute concentrations of only one factor will impact rel. contributions of all factors.**

→ We advocate for the use of relative contributions primarily because they allow a comparison among different sources without being influenced by overall pollution levels. In Fairbanks, the total pollution levels were strongly influenced by temperature and meteorology. Using relative contributions enabled us to overcome the effects of both total pollution and meteorology to divulge the changes in heating practices.

**Figure 1 B) Personally I find it hard to make a distinction between WS and WD colours on the graph. How was J(NO2) calculated? I do not think that is mentioned anywhere. E) I do not think Fig1E is discussed anywhere in the text. How is PM2.5 number concentration measured?**

→ jNO2 values were measured using a filter radiometer (MetCon Gmbh) installed at the NCore site. These measurements were not a part of our study. The original study reporting them has now been cited in the caption of Figure 1 (Simpson et al., 2024, https://doi.org/10.1021/acsestair.3c00076).

As for $PM_{2.5}$ measurement, there was a miscommunication on our part. Figure 1 does not show any PM2.5. The caption has been updated accordingly.

**Figure 2: if data points are sized, then size legend should be included.**

→ I acknowledge the reviewer's concern for this figure. Please note that the size of the data points is a rather trivial (or even redundant) aspect of this figure, so its explanation was shifted to the caption to avoid overcrowding the figure itself with legends. Data point 'sizing' in this figure is only meant to better visualise overlaying data points, which would otherwise completely cover one another. Keeping this in mind, we prefer to maintain the figure as it is and mention both the largest and the smallest sizes in the caption for simplicity.

**All mass spectra figures: considering CHARON resolution of around 5000, not more than 2 decimal places should be given for m/z.**

→ Thank you for your feedback. We completely agree and have revised Figure 4 accordingly.

**Figure 4 and 5: I suggest to combine these two figures to look the same as Fig 3 and put the last column in figure 5 in the supplement.**

→ This is actually an excellent suggestion and something that we did consider before submission as well. However, we would like to emphasize Figure 4 needs to be presented independently to accommodate all the essential ions' labels without crowding the figure.

**Figure 7A: on y-axis next to HR-ToF AMS put the ion in question**

→ Thank you for pointing out this error. It was a serious oversight on my part. I have corrected Figure 7 accordingly.

**Table 1: I suggest moving it in the supplement**

→ We have heavily relied on the information provided in **Table 1** to justify the identification of our factors. Therefore, we would like to keep it in the main text, even if it is toward the end.

**Supplementary:**

**S3: I am assuming the sentence " This introduces a sust the campaign, and so must the EF" is there by mistake**

→ Thank you for noticing this error. The sentence did not belong there and has been removed in the revised version.

**S5 and S6: Can you please briefly justify the number of bootstrap replicates for both PTR(CHARON) and AMS factorisation?**

→ The goal of bootstrapping in PMF is primarily statistical, helping in assessing the stability and uncertainty of source profiles through resampling the data. Some studies have used up to 250–1000 bootstraps runs (Tobler et al: https://doi.org/10.5194/acp-21-14893-2021; Chazeau et al. https://doi.org/10.1016/j.aeaoa.2022.100176), but we chose 100 because they provided a sufficient estimate of the stability of our solutions and uncertainty without being computationally demanding, given the large dimensions of all three of our datasets. Although using a larger number of bootstraps would undoubtedly yield even greater precision (e.g., reduced scaled residuals) and enhanced stability (e.g., same factor identities and distribution), we simply lacked the CPU/GPU power to have that many bootstrap runs. Particularly, since the residuals – both with and without bootstrapping – remained minimal, and we observed little to no change in the factor types identified, therefore, using a smaller number of bootstraps is justified.

**Figure S3: Why is not starting from 15nm?**

→ Thank you for noticing this oddity. The actual data from SMPS begins at 15 nm and we have revised Figure S3 accordingly.

**Figure S5: why should scaled residuals ideally be +/- 2?**

→ Scaled residuals are a metric to quantify how well the PMF model fits the provided data. Mathematically, it is described as follows: $(Observed\ Value - Modeled\ Value)/\ Uncertainty$. In PMF, it is generally recommended that scaled residuals remain within ±3 or 2, based on standard statistical principles (simply put, it is a glorified standard deviation!) for assessing model performance (EPA PMF User guide: https://www.epa.gov/sites/default/files/2015-02/documents/pmf_5.0_user_guide.pdf). Large positive scaled residual values suggest that PMF is not adequately fitting the input variables or that those variables are associated with an infrequent source. A good fit maintains a balance of accuracy and uncertainty, ensuring that most data points fall within this range without excessive outliers.

**General comment: the font in most of the figures (axis numbers, labels, legend) is too small. Please consider increasing the font.**

→ Thank you for the suggestion. We have enlarged the font in the axes, labels, and legends in all the figures in the main text. This has been done specifically for Figures 3, 4, 5, and 6.

Response to **Reviewer 2**

**The manuscript titled "Complementary aerosol mass spectrometry elucidates sources of wintertime sub-micron particle pollution in Fairbanks, Alaska, during ALPACA 2022 " uses aerosol mass spectrometry techniques, including CHARON PTR-ToF MS and HR-AMS, to analyze non-refractory particulate matter and its sources. Positive Matrix Factorization was applied to identify and estimate the contributions of various pollution sources, with a particular focus on residential heating emissions, on-road traffic, and secondary aerosol formation. The findings highlight the significant role of residential heating, alongside secondary organic aerosol processes, notably organosulphur, on air quality.**

**These observations offer valuable insights into this subarctic region, which is prone to significant particulate pollution events during winter. However, I have several concerns regarding the manuscript's structure, novelty, and methodology, which I outline in the general comments. A list of specific comments follows thereafter.**

→ Thank you for your thorough review of our manuscript. We value the time and effort you took to evaluate our research on wintertime sub-micron particle pollution in Fairbanks, Alaska. We appreciate your acknowledgement of the study's valuable insights into subarctic particulate pollution. Your feedback helped us improve the quality of our study.

**Structure and presentation:**

**1. The text is at times lacking conciseness and its structure could be optimized. In some instances, a potential issue is identified, but its possible causes are only discussed several paragraphs later, making the argumentation harder to follow. For example, the relatively low CHARON_traffic mass mentioned in L.486 is its potential causes debated only a few paragraphs later in the text, on the following page.**

→ We acknowledge the reviewer's concerns regarding the paper's structure and writing quality. In response, we have made significant revisions of whole article (abstract, results and conclusion). Notably, we separated the discussions on HOA and COA into two distinct sections (Sections 3.2.3 and 3.2.4). We have also simplified the explanations for the discrepancies in mass concentrations from the $PTR_{CHARON}$ and AMS by placing them in their respective sections, rather than placing it all the way at the end of both sections as originally done. And we reduced some sections. All changes are visible (track changes) but since too many changes have been realized to facilitate the reader, we also provide a revised article (no track changes)

**2. The inclusion of three PMF analyses for two instruments is not easy to follow, particularly due to the way the acronyms are introduced. The results section lacks clarity, as multiple PMF analyses are presented in a fragmented manner. Moreover, the added value of applying PMF to both AMS OA and OA+Inorganics is unclear. Upon reading, one has the feeling that PMF AMS organics could be removed from the manuscript without any meaningful loss while improving readability.**

→ We recognize that discussing three datasets can complicate readability. Considering that AMS is established as a refence technique, provides additional information on the inorganic fraction, we think it is important to keep the 3 PMF in this paper. We have streamlined the text by minimising the repetitive mention of $AMS_{org}$ and $AMS_{org+inorg}$, using just AMS where both are applicable. HOA and COA were also divided up into two sections. Additionally, we introduced some headers and specified a new section, "Contribution of Sulphate to the OOA Factor," to clarify the discussion on OOA.

**3. I don't find the figures particularly suitable for publication in their current form. Their layout, font sizes and colour choices should be refined to meet article-level standards.**

→ We have enlarged the font in the axes, labels, and legends in all the figures in the main text. This has been done specifically for Figures 3, 4, 5, and 6.

**Novelty**

**4. The authors repeatedly justify in the abstract and introduction that "significant uncertainties exist about aerosol sources, formation, and chemical processes during cold winter" in Fairbanks, Alaska (e.g. L.31). However, an extensive body of literature spanning over a decade has already attributed residential heating as a major driver of air quality degradation in that region, as well studies focusing on sulphur-containing particles. The manuscript should more precisely articulate the specific knowledge gaps that remain unresolved. Conversely, care must be taken not to extrapolate its findings to other arctic regions given the localized representativeness of their findings (e.g. L.138).**

→ We appreciate your point regarding the existing literature on residential heating as a significant contributor to air quality degradation in Fairbanks, Alaska. We acknowledge that previous studies have identified this factor, and we aim to build upon that foundation. **The novelty of this study is related to the added value of a novel soft-ionisation technique (CHARON PTR-ToF MS) in providing molecular-level insights together with high temporal resolution. This allows us to separate different types of biomasses burning organic aerosol (BBOA) and identify an oil heating factor, which has not been extensively characterised in prior research. We also computed a combined PMF with SMPS data in order to combine chemical factors with particle size distribution. This work enhances our understanding of the sources of organic aerosol during cold winter months in Fairbanks.**

**We modified or removed some of the sentences (L31 and L138 of the main text) and we rewrote the abstract.**

We recognise the importance of not extrapolating our findings to other Arctic regions without caution. Our results are indeed localised to Fairbanks, but they could be relevant for other urbanized sub-Arctic areas. So we have rephrased to "**fast-urbanising sub-arctic" and "anthropogenically-influenced regions of the sub-arctic"** to align with the reviewer's suggestion.

**5a. There appears to be little "complementarity" between CHARON and HR-AMS analyses in providing new insights into wintertime PM levels. CHARON independently proposes a novel separation of residential heating sources (discussed further below), while HR-AMS produces results that are largely consistent with prior literature (traffic, residential heating, and OS as key winter**

**contributors). Some of these findings have already been reported in studies from related or the same field campaigns (e.g., Campbell et al., 2022; Robinson et al., 2024; Yang et al., 2024).**

→ The two deployed instruments are “objectively” complementary – even without the context of our study. One ensures very good mass closure, identification/quantification of both an organic and inorganic fraction but with reduced capability for compound identification, while the other delivers significantly better source resolution for the organic fraction. Together, they enhance our understanding PM composition and sources.

The reviewer rightly cites the contributions of Campbell et al. (2022), Robinson et al. (2024), Yang et al. (2024), Edwards et al. (2024), and Campbell et al. (2024) regarding air quality dynamics during the ALPCA campaign in Fairbanks. While overlaps in findings are common in scientific research (especially during a field campaign), our paper offers additional insights, particularly in delineating $PM_1$ sources and moving beyond single-instrument studies.

Our focus on the chemical composition of aerosols – particularly $PM_1$ – addresses significant health concerns and underscores the necessity for further investigation and regulation of specific source contributions. By identifying distinct types of wood and oil burning in residential heating, we aid targeted mitigation strategies. Additionally, our examination of resident behavior in response to burn bans contributes to understanding policy effectiveness. The molecular-level information we provide is fundamental for assessing potential health impacts, including evaluating factors, such as total PAHs and their associated toxicity.

For comparison, the following table outlines other closely related studies conducted during the ALPCA campaign, including those mentioned by the reviewer:

| | |
|---|---|
| **Edwards et al., 2024: Residential Wood Burning and Vehicle Emissions as Major Sources of Environmentally Persistent Free Radicals in Fairbanks, Alaska** | First study to quantify EPFRs in Fairbanks and link them to specific sources and health risks (particularly oxidative stress). |
| **Campbell et al., 2024: Enhanced aqueous formation and neutralization of fine atmospheric particles driven by extreme cold** | Investigates the unique role of low temperature in enhancing aqueous reactions and modulating secondary aerosol formation, particularly HMS, through its impact on particle pH. |
| **Yang et al., 2024: Assessing the Oxidative Potential of Outdoor PM2.5 in Wintertime Fairbanks, Alaska** | First assessment of oxidative potential (OP) in Fairbanks $PM_{2.5}$, contrasting it with other urban areas and linking OP to specific sources. |
| **Robinson et al., 2024: Multi-year, high-time resolution aerosol chemical composition and mass measurements from Fairbanks, Alaska** | Provides a unique, multi-year, high-time resolution dataset, offering a comprehensive overview of aerosol behavior in Fairbanks across different seasons and meteorological conditions using long-term ACSM data. |
| **Campbell et al., 2022: Source and Chemistry of Hydroxymethanesulfonate (HMS) in Fairbanks, Alaska** | First in situ measurements of HMS in Fairbanks, highlighting its significant contribution to $PM_{2.5}$ sulfur and exploring its relationship with meteorological parameters and other species. |

**Given the study's title, I expected some type of combined analysis of CHARON and HR-AMS to yield results beyond the simple sum of their individual analysis, but this expectation is not met.**

The title has been modified to "***Sources of wintertime sub-micron particle pollution in Fairbanks, Alaska, during ALPACA 2022"***

→ We chose to analyze CHARON PTR-ToF MS and HR-ToF AMS datasets separately, a decision that may seem counterintuitive given the value of combined factor analysis, as reported by Tong et al. (2022) (doi: 10.5194/amt-15-7265-2022), among others. However, this choice was driven by the preliminary idea to work preferentially on the $PTR_{CHARON}$ dataset. By applying traditional PMF to $PTR_{CHARON}$ and using HR-ToF AMS as a validation tool, we successfully met our primary goals of identifying $PM_1$ sources in Fairbanks and highlighting the $PTR_{CHARON}$ capability to help do so. We also decided to change the title of the article as indicated above.

**Methodology:**

**6. The PMF on CHARON data is difficult to verify, notably with this complex 8-factor solution. As far as I can gather, I am not completely convinced that the four residential heating factors are well-separated. This is often an issue for co-varying factors, which is the case here where Pearson is between 0.6 and 0.75.**

**Furthermore, there seems to be some issues among the four residential heating factors. For example, when normalized to their OA contributions, ResH2 (attributed here to hardwood combustion) and ResH3 (attributed to heating oil combustion) have the same relative fraction of levoglucosan - a compound that is a unique tracer of cellulose pyrolysis.**

→ We understand the reviewer's concerns about the complexity of the 8-factor solution and the potential for co-varying factors. However, we would like to assert that we have thoroughly evaluated the PMF results and molecular compositions to separate residential heating sources accurately. Table S2 presents a summary of PMF results from 3 to 10 factors. In the $PTR_{CHARON}$ a splitting is observed for more than 8 factors, while below 8 factors, we could see mixing of ResH4 (hardwood) and ResH3 (heating oil), which are different sources based on their molecular compositions and correlation with $SO_2$. Furthermore, when combining OA PMF-charon with size distribution data, we see that the factors are associated with particles having distinct size distributions (notably ReshH3 is associated to smaller particles than the other wood combustion factors, see Figure S12 plot A). This supports our choice of an 8-factor solution as the most stable representation of the data.

Regarding the separation of the four residential factors, in the specific case of Fairbanks we rather rely on the molecular composition than on time series as the latter are expected to strongly co-vary due to the co-emissions during strong temperature inversions. Also the time correlation of the 4 residential factors is not as good as we could expect for split factors ($R^2$ from 0.35 to 0.55) as can be observed in the table below.

| A | Traffic | COA | OOA | RH1 | RH2 | RH3 | RH4 | SM |
|---|---|---|---|---|---|---|---|---|
| Traffic | 1.00 | | | | | | | |
| COA | **0.29**
a=0.04 ± 0.00
b=1.20 ± 0.01 | 1.00 | | | | | | |
| OOA | **0.17**
a=0.17 ± 0.00
b=3.42 ± 0.06 | **0.11**
a=0.25 ± 0.00
b=1.24 ± 0.03 | 1.00 | | | | | |
| RH1 | **0.27**
a=0.29 ± 0.00
b=3.55 ± 0.04 | **0.19**
a=0.38 ± 0.00
b=1.33 ± 0.02 | **0.18**
a=0.40 ± 0.00
b=0.34 ± 0.00 | 1.00 | | | | |
| RH2 | **0.07**
a=0.58 ± 0.02
b=6.87 ± 0.20 | **0.08**
a=0.67 ± 0.02
b=3.22 ± 0.09 | **0.15**
a=0.57 ± 0.02
b=1.16 ± 0.02 | **0.36**
a=-0.23 ± 0.02
b=2.30 ± 0.02 | 1.00 | | | |
| RH3 | **0.10**
a=0.44 ± 0.01
b=2.81 ± 0.06 | **0.12**
a=0.48 ± 0.01
b=1.28 ± 0.03 | **0.12**
a=0.49 ± 0.01
b=0.36 ± 0.01 | **0.52**
a=0.1 ± 0.00
b=0.94 ± 0.01 | **0.41**
a=0.43 ± 0.00
b=0.21 ± 0.00 | 1.00 | | |
| RH4 | **0.08**
a=0.50 ± 0.01
b=3.48 ± 0.10 | **0.11**
a=0.52 ± 0.01
b=1.74 ± 0.04 | **0.06**
a=0.61 ± 0.01
b=0.34 ± 0.01 | **0.35**
a=0.09 ± 0.01
b=1.08 ± 0.01 | **0.50**
a=0.39 ± 0.01
b=0.32 ± 0.00 | **0.55**
a=0.03 ± 0.01
b=1.05 ± 0.01 | 1.00 | |
| SM | **0.06**
a=0.16 ± 0.00
b=0.47 ± 0.01 | **0.05**
a=0.29 ± 0.00
b=3.55 ± 0.04 | **0.02**
a=0.18 ± 0.00
b=0.03 ± 0.00 | **0.00**
a=0.19 ± 0.00
b=0.00 ± 0.00 | **0.01**
a=0.18 ± 0.00
b=0.01 ± 0.00 | **0.00**
a=0.19 ± 0.00
b=-0.00 ± 0.00 | **0.01**
a=0.66 ± 0.01
b=0.51 ± 0.06 | **1.00** |

Regarding the relative fractions of levoglucosan in ResH2 (hardwood combustion) and ResH3 (heating oil combustion), there may be some misunderstanding. Approximately 30%, 14%, 9%, and 26% of the protonated levoglucosan signal and its fragments are distributed across ResH1, ResH2, ResH3 and ResH4, respectively, indicating an uneven distribution of this tracer through the 4 factors. While PMF has inherent complexities, our results suggest clear distinctions in levoglucosan allocation across the four residential heating factors.

**Furthermore, all four factors exhibit a fairly high and comparable correlation with $SO_2$, while ResH1 "mixed wood burning" even shows a stronger correlation with "sulphate" than heating oil combustion, discussed here and previous references to be the main source of sulphur dioxide. I'll refrain from commenting on the separation between softwood vs. hardwood emissions, i.e. factors ResH2 and ResH4, as I am not an expert on the topic, but it also seems to be at least somewhat debatable. I believe this needs to be strengthened. As I discussed above, the separation of residential heating sources is a key novelty aspect of this work.**

→ We appreciate the reviewer's comment regarding the correlation between residential heating factors and $SO_2$. We would like to highlight that the correlation coefficients for each residential heating factor with $SO_2$ present some difference. Specifically, ResH1 (mixed BB), ResH2 (hardwood), and ResH4 (pinewood) show correlation coefficients of 0.46, 0.34, and 0.47, respectively, compared to 0.61 for ResH3 (heating oil combustion). While one could argue that the $R^2$ differences are insufficient to attribute $SO_2$ solely to ResH3, as mentioned above particle size distribution supports the interpretation of a combustion oil factor associate to small particles while the other factors are rather related to larger particles (Figure S12 plot A).

PMF is inherently uncertain, often relying on informed interpretations based on time series data (e.g., rush hours for traffic, week-day or week-end, ect). We provided a related disclosure in the manuscript: "***The specific nature of wood cannot be inferred unambiguously, as emissions of marker species like levoglucosan or methoxy phenols vary with the type of fuel, heating appliance, operational conditions, and combustion cycle stage (Fine et al., 2002; Alves et al., 2017). Nevertheless, several studies (Fine et al., 2002; Schauer and Cass, 2000; Kawamoto, 2017) have differentiated between softwood and hardwood by identifying marker compounds present in our study, such as substituted phenols and resin acids*".** We minimised guesswork by relying on well-established marker species from the literature. Our identification of hardwood and pinewood species is grounded in objective analysis of these marker species, detailed in Section 3.2.2. We are confident in our 8-factor solution, evidenced by clear factor

separation at higher numbers and mixing at lower numbers, as well as the identities assigned to each source. However, we are open to revising our interpretations if the reviewer could clarify/specify why the hardwood and pinewood identities should not be trusted.

**7. L.216-219: It's unclear why m/z > 120 was used for OA calculation but removed from PMF analysis, notably when co-emission of PAH is expected with factors such as BBOA and HOA. Their use as "external tracers" raises some questions about the robustness of the analysis. Please provide arguments to justify this (as well as re-phrase sentence for clarity).**

→ Our initial analysis incorporated all PAHs in the AMS analysis, and the resulting factorisation was the same as what we presented in the paper, achieving the primary aim. Another possible concern regarding the removal of PAHs is the potential underestimation of the mass concentration attributed to the factors. However, ~98% of AMS signal was present in fragments with *m/z* <120. Still, this limitation was acknowledged in the original manuscript as follows (line 251): *"**This exclusion is expected to cause underestimation (by <2%) of the mass of some factors, particularly HOA (hydrocarbon-like organic aerosol) and BBOA (biomass-burning organic aerosol)".***

Furthermore, it is quite common to restrict the PMF on the high-resolution organic mass spectrum to *m/z* 120 (e.g., Aiken et al, https://doi.org/10.5194/acp-9-6633-2009; Young et al, https://doi.org/10.5194/acp-16-5427-2016, Duan et al, https://doi.org/10.5194/acp-22-10139-2022; Qi et al., https://doi.org/10.1016/j.scitotenv.2021.151800; among many others). This approach is warranted because most of the signal from OA and inorganic species is produced within this mass range, largely due to extensive fragmentation resulting from electronic impact ionization of larger molecules. So finally, we extracted PAHs from AMS measurements to utilise them as external tracers (Figure 5). Specifically, we used them to differentiate factors like BBOA and HOA from other factors.

**Specific comments**

**1. Please rework the abstract providing a more quantitative view of the results presented here, highlighting their novelty.**

à We have made substantial modifications to the abstract, the updated text is as follows: ***"Fairbanks, Alaska, is a sub-arctic city that frequently suffers from non-attainment of national air quality standards in the wintertime due to the coincidence of weak atmospheric dispersion and increased local emissions. As part of the Alaskan Layered Pollution and Chemical Analysis (ALPACA) campaign, we deployed a Chemical Analysis of Aerosol Online (CHARON) inlet coupled with a proton transfer reaction - time of flight mass spectrometer (PTR-ToF MS) and an Aerodyne high-resolution aerosol mass spectrometer (AMS) to measure organic aerosol (OA) and NR-PM$_1$, respectively. Positive Matrix Factorisation (PMF) analysis was used for source identification of the NR-PM$_1$. The PTR$_{CHARON}$ factorisation of the organic fraction identified four residential heating sources, with oil combustion accounting for 16 % and wood combustion contributing significantly at 47 %. The analysis could further differentiate between hardwood and softwood combustion. The AMS analysis revealed only one biomass burning-related factor contributing 28 % of total OA and assigned the largest organic fraction to a hydrocarbon-like factor related to road transport with 38 % of the OA. Additionally, two factors enriched in sulphate (including both organic and inorganic fraction) and nitrate were identified. These***

***results evidence the complementarity of the two instruments and demonstrate as $PTR_{CHARON}$ provides both qualitative and quantitative information, offering a comprehensive understanding of the organic aerosol sources. Such insights into the sources of sub-micron aerosol pollution can assist environmental regulators and citizen efforts to improve air quality in Fairbanks and the fast-urbanising regional sub-Arctic areas."***

**2. L. 35: "Which" instead of "that".**

→ we have modified the section.

**3. L101: remove "in detail"**

→ The phrase, 'in detail,' was an unnecessary descriptor and has been removed as you suggested.

**4. L111: Do you mean "more selective"? Or just selective, since it depends on proton affinity.**

→ Selectivity – less or more – was irrelevant to the intended message, so it has been omitted for brevity.

**5. L.120: "Information on NR-PM1 OA" or something along those lines.**

→ We have rephrased this sentence to: ***" the former features molecular level information of the OA faction but has limited ability to detect particles below 150 nm (Eichler et al., 2015); the latter covers smaller particle size range (i.e., > 60 nm) and detects inorganic components too (Decarlo et al., 2006). Together they provide an excellent combination of real-time and quantitative data on atmospheric ambient aerosol."***

**6. L.127: "Not well understood".**

→ The suggested change has been made: ***"The detailed composition of sub-micron aerosol in Fairbanks and other anthropogenically influenced sub-Arctic regions – is not well-understood"***

**7. L134: I'd advise maintaining sulphate, or rewrite "chloride-, nitrogen- and sulphur-containing species". Albeit well known by the community (and underlying current understanding of sulphate at Fairbanks), sulphur is the element, not the species or the aerosol type.**

→ We agree with the reviewer. Our reasoning behind using "sulphur," rather than "sulphate" was indeed the possibility of the contribution from non-$SO_4^{2-}$ species. We have revised in the entire manuscript.

**8. L.134: "What does "good mass resolution" mean? ToF-ACSM would be enough? V-mode? W-mode? I suggest being more accurate here, notably on the role of mass resolution in the findings of the manuscript.**

→ We have revised the whole text we remove resolution the actual version is the following ***:***

*" These findings highlight the synergistic benefits of combining multiple analytical techniques and emphasise how soft ionisation mass spectroscopic methods enhance molecular-level insights into particulate organic carbon. This integrated approach advances our understanding of the complex composition of particulate matter, offering valuable contributions to environmental characterisation and source apportionment studies."*

**9. L.150: "recorded"**

→ The tense has now been corrected (now line 149).

**10. L.182: What is the validity of Leglise fragmentation correction into Fairbanks OA?**

→ The Leglise et al. method addresses a key limitation in $PTR_{CHARON}$ mass spectrometry: the fragmentation of analyte ions during PTR ionization can introduce a negative bias in quantification. This correction is particularly important for comparison between $PTR_{CHARON}$ and AMS measurements of OA. Its application does not overcome the underestimation of vehicle emissions (due to the bad detection of aliphatic compounds as alkanes, etc.) known to undergo dissociative PTR ionisation. Despite applying this correction, there remains some underestimation in $PTR_{CHARON}$ measurements of OA. The exact uncertainty associated with this should be explored via controlled laboratory experiments on known OA mixtures in the future.

**L.199: RIE for sulphate was 1.93. Is that consistent with previous characterizations of the instrument? Has estimated SO4 from HR-AMS been compared with other observations?**

→ Yes, typically an RIE within the 1.8–1.9 range is obtained for sulphate using this instrument. The sulphate levels estimated with the AMS were found to be 30% lower than those obtained from IC measurements from $PM_{0.7}$ filter samples. Additionally, the nitrate and ammonium components determined with the AMS also showed almost 30% reduction compared to the IC measurements, suggesting that such systematic differences among all species could rather arise from sampling uncertainties and/or non-overlapping particle sizes.

**11. L.203: Confirm that the CDCE algorithm on PIKA calculated a correction factor down to 0.35**

→ The correction factor of 0.35 is the experimental CE observed when the ammonium concentration was below the threshold of 0.03 $\mu g/m^3$. This value was used as the default CE input for the CDCE algorithm.

**12. L.237: please develop "to understand the data" for clarity.**

→ We have now updated this sentence as follows: ***"Preliminary PMF was performed without using a priori information to explore the dataset's structure, potential factor variability, and source contributions and guide the selection of an optimal solution before applying constraints."***

**13. L.239-242: Re-phrase for improved readability.**

→ We have considerably shortened and simplified the language as follows :

*"**We considered solutions from 3 to 13 factors, applying a step-wise, cell-wise down-weighting approach: variables with S/N < 0.2 ("bad" variables) were down-weighted by a factor of 10, while those with 0.2 < S/N < 2 ("weak" variables) were down-weighted by a factor of 2 (Paatero and Hopke, 2003; Ulbrich et al., 2009). Once primary factors, such as cooking and biomass burning, were successfully identified in unconstrained trials, we narrowed the range of possible solutions by applying the a-value approach, which allows for improved factorisation by constraining the PMF with external data when available (Canonaco et al., 2013; Paatero, 1999).**"*

**14. L.272: the correct SI notation for the second is "s" and not "sec".**

→ Thank you for pointing out this error. We have converted 'sec' to 's' throughout the revised manuscript.

**15. L.274: Indicate what the variability range stands for.**

→ The average values of BC and NR-$PM_1$ measured with the MAAP and AMS were **1.4 ± 1.4 µg/$m^3$** and **8.3 ± 9.3 µg/$m^3$**, respectively (variability range stands for standard deviation over the entire field campaign).

**16. L.276: remove "campaign".**

→ We removed campaign average

**17. L.276-L.278: Methodology section, on ancillary observations?**

**18.** → Ancillary observations in this study included SMPS, CPC, and OPC using standard protocols. These were specified in Section 2.1 of the Methodology (now highlighted in yellow).**L.291-298: I find this paragraph somewhat unclear, albeit highly relevant. Is 9% of the CHARON mass attributed to heteroatomic ions, being roughly 7% oxygenated ions (CHO) and the rest ON and OS? Are all OS and ON removed from the source apportionment analysis? Also, it's unclear what the relevancy of those "prominent peaks" is, they don't seem to be identifiable in Figure S8.**

→ 9% of the OA mass from $PTR_{CHARON}$ is attributed to heteroatomic species, that include organo-nitrates (ON) and organosulphates (OS). We believe there is a misunderstanding regarding the "7% of oxygenated ions (CHO)". We explicitly state that "***Detailed molecular-level composition of organics with the $PTR_{CHARON}$ reveals a large majority of organics to comprise only C, H, and/or O atoms, while only ~9% of the $OA_{CHARON}$ mass measured with this instrument was attributable to heteroatomic species, including organonitrates and organosulphates (Figure S8)***."

The OS and ON components were retained for source apportionment, although we did not dwell on for "factor identification". Their distribution can be seen in Figure 4. It is true that the ON and OS peaks are not discernible in Figure S8, which shows the contributions of all OS, ON, and CHO peaks combined. We have now also referred to Figure 4 in this text, which distinctly shows these peaks. Regardless, you are justified in questioning the "relevance" of these peaks in this context. They were mentioned for the benefit of interested readers – to indicate their presence, even though we did not elaborate on them extensively due to the scope of this particular work.

19. L.278-281: eBC has already been estimated at 15% of PM1, how can PM2.5 be 99% of NR-PM1? There is a quantification issue with one (or both) methods, besides other refractory PM1 species that have not been considered, and the PM1-PM2.5 fraction. The analysis seems to completely miss those basic considerations.

→ Thanks for catching this error, the sentence was confusing, we deployed an OPC at the site and comparing PM1 and PM2.5 we can say that PM1/PM2.5 is 0.99. Now this reads in the manuscript "***Ancillary measurement at the CTC site with an OPC showed that the hourly PM1 comprised up to 99% of the PM2.5***"

**20. L.304: avoid "could unequivocally be identified"**

→ The term, unequivocally, has been removed from the sentence.

**21. Figure 2: To improve readability I suggest not to colour code against PMF factors that have not yet been presented in the manuscript but some standard tracers (whether levoglucosan from CHARON or more "usual" tracers from AMS like f60 and f55, for example). Otherwise, panes D and E can go into supplementary, where results from different sections can be combined without impacting the flow of the manuscript.**

→ Figure 2 has now been updated to remove panels D and E. The latter two panels are now a part of Figure S9.

**22. L.308-312: Shouldn't the size-dependent EF correct for that effect?**

Yes, the size-dependent EF is meant to correct for that effect. According to the results, the correction has only a limited compensatory effect for particle below 100nm. Therefore, sources that produce ultrafine particles (≤100nm), including vehicular emissions and fuel oil, are the most affected.

**23. L.316-318: Has total OA been observed to decrease during laboratory experiments in the lab or just the concentration of those particular species? Fragmentation does not necessarily induce OA mass loss, it will of course depend on the PA of the fragments.**

→ Equation S2 clearly shows that OA concentrations can be biased toward molecules yielding fragments with *m/z* values that are significantly lower than their molecular weight. This was the case for alkanes C16–C26 alkanes that we tested in the laboratory which extensively fragment into $C_nH_{2n+1}$ with the most abundant ions being *m/z* 43, *m/z* 57, *m/z* 71, and *m/z* 85. Taking hexacosane as an example, assigning the original molecular weight (i.e $m/z_i$ = 366) to each of the fragments resulted in a mass concentration that was roughly 4 times higher than the one obtained using fragments' individual *m/z*.

**24. L.335: Clarify PMF of OA and OA+inorganics. Given that OA+inorganics analysis has seen added one factor for NO3 and one for SO4, it's not unexpected that the main source of OA has remained unchanged.**

→ We appreciate the reviewer's suggestion. Indeed, the distinction between the PMF of $AMS_{org}$ and $AMS_{org+inorg}$ only lies in the additional factors, ammonium nitrate (AmNi) and sulphate-rich OA (Sulph-OA). As expected, the primary OA sources remain mostly unchanged. Nevertheless, we conducted a first separate $AMS_{org}$ PMF for direct comparison with PMF of $PTR_{CHARON}$.

**L.346: It's curious about the low correlation with BC for such a dominating OA source, has it been corroborated also on previous studies for this site? Is there an explanation for the complete lack of correlation with CO?**

→ The lower correlation between BC and the dominant OA source (i.e., residential heating) may result from differences in the combustion conditions. Wood-burning emissions can be very variable, producing organic-rich emissions with varying BC content depending on the burn phase and type of appliance used for burning. Previous studies in Fairbanks have similarly reported weak or inconsistent correlations between BC and OA from wood combustion. For example, Ward et al. (2012) observed that while wood combustion is a significant $PM_{2.5}$ source, the BC-to-OA ratio varies greatly with the combustion conditions. Similarly, Robinson et al. (2023) reported that BC and OA from wood burning do not always correlate strongly, which could be due to differences in combustion efficiency and meteorological conditions. These studies support our observation of BC not being a reliable tracer for BBOA, especially from residential sources, in Fairbanks.

As for CO, the CTC site being located close to main roads, it is expected that the CO variations are driven by the vehicle emissions. This is confirmed by the measured diurnal cycle of CO that follows the on-road mobile sector on the sampling site, as shown by Brett et al (2024) (Figure 9A in Brett et al. 2025; https://doi.org/10.5194/acp-25-1063-2025.

**25. L.374: Rephrase the analysis (also L.369), where 14% (ResH2) of LEV is considered robust and 9% (ResH3) minor.**

→ We would like to assert that the comparison of levoglucosan (LEV) contributions to the different residential heating factors could have been misunderstood to some extent. While ResH2 does contain 14% of the total levoglucosan signal and is considered a major wood-burning factor, ResH3 at 9% is still a relevant source but is more strongly associated with oil combustion due to its distinct molecular signature, containing enhanced quantities of PAHs and a better correlation with gaseous $SO_2$. The text has been revised as follows:

"***Levoglucosan is used here as an internal tracer of biomass burning being relatively stable under atmospheric conditions (Fraser and Lakshmanan, 2000). Most of signal from protonated levoglucosan (m/z 163) and its fragments (at m/z 85, 127, and 145) were found in ResH1, ResH4, and ResH2 (13-29%, Figure S11), while ResH3 contained a smaller fraction (i.e., 9%), suggesting that at least three factors do originate from biomass burning – more specifically,***

***wood-burning (Figure 4 and S11). These three wood-burning related factors collectively accounted for an average of 2.1 ± 2.5 µg/m³ of $OA_{CHARON}$ (corresponding to 47 ± 20% of total factorised $OA_{CHARON}$).”***

L.412: present the relative contribution in the form of an equation.

This formula has been used to calculate and has been added to supplementary section:

$$RI_{i,f} = \frac{C_{i,f}^{norm} - C_{i}^{norm}}{\sigma_{i}^{norm}}$$

Where:
$RI_{i,f.}$ = the relative importance of variable *i* in factor *f*
$C_{i,f}^{norm}$ = normalised concentration of variable *i* in factor *f*
$C_{i}^{norm}$ = mean normalised concentration of variable *i* across all factors
$\sigma_{i}^{norm}$ = standard deviation of the normalised concentration of variable *i* across all factors

**26. L.460: The journal and DOI of this reference are lacking, I could not find it.**

à This reference pertains to a poster from our group on this work presented at the EGU, so there is no DOI. We have provided the link to the abstract:
https://agu.confex.com/agu/fm22/meetingapp.cgi/Paper/1072876

**27. L.466: Please include already here possible explanations for discrepancies on mass loadings between HOA from AMS and traffic factor from CHARON from page 14.**

→ We thank you for pointing out this disorganised arrangement of text. We have now created two distinct sections for HOA and COA, separately addressing the reasons for the discrepancies in their mass concentrations from $PTR_{CHARON}$ and AMS, rather than combining them at the end. More details are discussion in the SI section.

**28. L.475-477: Rephrase “reliable tracer for it is yet to be identified” as a direct sentence.**

→ Rephrased to “***Although a reliable tracer for COA remains unidentified, a high f55/f57 ratio of >1 is considered characteristic (Katz et al., 2021; Sun et al., 2011)***.”

**29. S3: correct “sust”**

→ Thank you for catching this error. This entire phrase was mistakenly left in the supplement and has now been removed.

**30. Fig. S3: Most of the EF calculated is above 7, which if I read correctly implies a volume-weighted distribution generally >200nm for submicrometric aerosols. Is that correct?**

→ Yes, you have read correctly! Such an EF value implies that a considerable "concentration-weighted" portion of the $PM_1$ OA mass was associated with particles with diameters ≥ 200 nm. The EF values > 7 indicate a shift toward accumulation-mode aerosols, which is consistent with emissions from biomass burning and aged aerosol. This also agrees with the SMPS data that showed that ResH1, ResH2 and Resh4 factors are dominated by particles in the 200–400 nm range (Figure S12 A).

Response to **Reviewer 3**

**The authors describe analysis of a mass spectrometry dataset collected in Fairbanks, Alaska, aimed at investigating sources of ambient $PM_1$ that contribute to local poor air quality events resulting in non-attainment of health-based air quality standards. Local meteorology, in particular temperature inversions, lead to accumulation of emissions from local activities including home heating, traffic and cooking. The novelty of this work lies in the additional insights obtained from leveraging the PTR-CHARON molecular compositional data. Most real-time organic aerosol apportionment studies rely on analysis of heavily fragmented EI mass spectra from AMS instruments. While these are useful for mass closure, and for informing air quality policy (eg traffic vs woodburning mass contributions) relatively little information on organic aerosol composition is obtained. In this work, separate, thorough analyses of the AMS organic, AMS organic+inorganic and the PTR-CHARON datasets through PMF are used to maximize the value of each technique. There are also synergistic aspects. For example, the CHARON dataset reveals multiple distinct local residential combustion source profiles featuring marker ions that can be associated with hardwood, softwood, resin and even oil combustion, whereas only a single residential combustion factor (BBOA) is extracted from the AMS data. The identification of specific marker ions (eg furfural, guiacol, eugenol, vanillin, coniferaldehyde) will undoubtedly be useful for CHARON users going forward. On the other hand, mass closure appears to be an issue for the CHARON dataset, which is limited by poor particle transmission at lower sizes and significant fragmentation of alkanes despite the soft ionization used. In the case of this work I don't believe the mass concentration underestimation of the CHARON for the various factors is too concerning because the AMS functions as a quantitative instrument for closing $PM_1$ OA. However, I think the conclusions would benefit from a broader discussion on where the authors feel that CHARON systems fit in to future source apportionment efforts that aim to instruct policy. Should an AMS system always be co-deployed for example, to ensure accurate OA mass loading measurements, with the emphasis on the CHARON data to speciate the AMS "factors" in more detail? Is there a potential for the CHARON data to target and reasonably accurately quantify specific air toxics that represent health risks, for example PAHs or quinones? Or does the size-dependent transmission efficiency introduce too much uncertainty in the absence of co-located SMPS measurements? This would be useful for the community. Overall, I find the manuscript to be well written and the analysis is rigorous, and I have only minor comments below.**

→ Thank you for your thoughtful review of our manuscript. We appreciate your acknowledgement of the novelty in leveraging PTR-CHARON data for enhanced organic aerosol compositional insights, complementing traditional AMS analysis. We agree that mass closure is a limitation, particularly for smaller particles and fragmented alkanes. As noted, the synergistic approach with AMS ensures quantitative $PM_1$ OA assessment. We have carefully addressed this point, as well as other concerns and suggestions, in the itemized response below. We have modified the conclusion, adding some discussion about the use of these two instruments. Due to quite severe requests from reviewer 2 the article has been modified abstract, some results section and conclusion have been rewritten. We therefore provide two version of the revised article (one with track changes, quite difficult to read and a second one without track changes).

**How often was zeroing performed for the CHARON mass spectral signals? Any humidity dependence of instrumental response?**

→ We specified in the manuscript (last paragraph of section 2.1.) as follows : ***"A HEPA filter was placed upstream of the inlet for an hour at regular intervals (twice a week) to measure the instrumental background."***

Extremely low temperatures were encountered during the campaign outside the trailer. The water content estimated using temperature and relative humidity from the gauge positioned at the AMS inlet was 1.2 ± 0.8 mmol/mol, corresponding to an equivalent relative humidity at 25°C of 2.65 ± 1.75% (trailer temperature was around 5-7 °C). Humidity was quite constant throughout the campaign in the inlet, and dependence is considered negligible in this range.

**For the PTR, for the species in the gas calibration mix, how well do the reported mixing ratios based on transmission efficiency and dipole moment/polarizability-derived k values agree with the expected mixing ratios?**

→ The table below lists the measured vs expected mixing ratio for the calibrated compounds.

| VOC | Measured/Expected |
|---|---|
| 33 | 0.97 |
| 42 | 1.06 |
| 45 | 1.25 |
| 59 | 1.16 |
| 69 | 0.84 |
| 79 | 0.91 |
| 93 | 0.96 |
| 107 | 0.96 |
| 121 | 0.97 |
| 135 | 1.04 |

The difference between the expected and measured mixing ratio arises only from using the modeled transmission curve. The *k* values do not affect the calculation of mixing ratios: since they are used to determine relative transmission, and the conversion from cps to ppb relies on the same *k* values, their influence is canceled out.

**Are the signals for BTEX observed in the constrained CHARON traffic factor from the condensed fraction of those species in OA or breakthrough of gases from the denuder?**

→ The combined effect of slight gas breakthrough from the denuder and memory effect from the previous VOC measurement could explain the presence of a residual signal (2-3% of the signal found during the gas-phase measurement) for these compounds in CHARON mode.

**If the residential heating factors co-vary how were they resolved by PMF? Do they have slightly different temporal dependencies? Presumably home heating emissions from all fuel types peak at the same times of day. Or is there a slightly different wind dependence that results in changes in the relative contributions to OA from these different fuel types because of different uses in different parts of the community.**

→ PMF is dependent on not just temporal trends. The residential heating factors co-varied but not a hundred percent. As shown in Table S5, $R^2$ values ranged between 0.35 and 0.52 for the four residential heating factors. This temporal difference, along with the presence of different molecular markers from the various fuel types, allows PMF to effectively resolve them.

**Figure 4: What is the SM factor? Small carboxylic acids? Is the acetone from denuder breakthrough? Does the lack of a diurnal trend mean this is background OA?**

→ The SM factor was an instrumental artefact, which was discussed in the supplementary section. I have appended this paragraph below for your review:

***Supplementary section. S5. Specifics of the factorisation of $PTR_{CHARON}$ measurements***

***"****The factorisation of $PTR_{CHARON}$ produced a unique factor that comprised largely of very small ions of m/z < 65, labelled as the small molecules (SM) factor. This factor could not be given an environmentally relevant identity based on a lack of correlations with the external tracers, and thus, it has not been discussed in the main text. Its major constituents were small species, such as m/z 59.05 ($C_3H_6O$), 61.03 ($C_2H_4O_2$), 73.03 ($C_3H_4O_2$), 75.04 ($C_3H_6O_2$), etc., that are expected to be in the gaseous phase, rather than the condensed phase. Its average concentration was* ***0.2 ± 0.1 µg/m³*** *with large relative contributions to total OA (>80%) during short and clean periods of the campaign, when ambient $OA_{CHARON}$ was below 1 µg/m³. We speculate that the SM factor is an artefact produced by instrumental chemical background and possible remnants of VOC species on the denuder as we switched from collecting gas-phase samples to particle sampling through the CHARON inlet****."***

**Was there an external validation of the AMS total mass concentrations from a gravimetric/nephelometer particle mass monitor?**

→ External validation of AMS total mass concentrations of $PM_1$ was conducted using mass concentrations from IC (PM0.7) of sulphates, nitrates, ammonium as shown in **Figure 7**.

**How is the fragmentation correction performed for the CHARON data? Are smaller fragment ions quantified or is there a fraction from the MH+ signal that is assumed to be "missing"?**

→ Fragmentation correction in our study does not involve direct quantification of small fragments or parent ions as we cannot differentiate between fragment and parent ions in a complex unknown aerosol mixture. Instead, we used correction factors derived in a previous study from the fragmentation patterns of 26 known oxidized organic compounds that were representative of atmospheric particulate organic matter (Leglise et al. 2019 https://doi.org/10.1021/acs.analchem.9b02949). The authors validated their correction methodology with the HR-ToF AMS measurements.

**The description of the varying enrichment factors is hard to follow. How is it dependent on concentration? Isn't the EF selected for each hour based on the mass-size mode of the SMPS? Although there seems to be bimodal distributions for some of the residential factors in the SI. Are the EF values relatively low compared to other studies?**

→ We are sorry that the reviewer found the EF-related discussion hard to follow. It seems there has been a misunderstanding. First, SMPS data was not collected hourly; all our data, including the concentration-weighted EF, depicted in Figure S3 has a 2-minute resolution. Second, the EF was not "selected". Rather, it was calculated at each sampling time point using the most predominant size as seen by the SMPS. How do we know which size is the *most predominant* at a given moment? - By picking the particle size

presenting the largest concentration. Hence, "concentration-weighted EF". The mathematical expression for this and associated discussion is given in Section S3 of the supplementary.

$$EF_{weighted} = \frac{\sum_{k=1}^{k=n} EF_k \cdot Conc_k}{\sum Conc} \text{……………}\textbf{Equation S3}$$

Yes, the EF values were much lower during the ALPACA campaign compared to those reported in other studies and the manufacturer, Ionicon, as shown in the Figure below.

[Figure]

**Figure 8 and the corresponding figures in the SI are very hard to follow. I think to show how the relative mass contributions of the residential factors change before, during and after an advisory period, the format in Figure 1D might be useful. If it's noisy the temporal resolution could be reduced. It's hard to track all of the box plots and in the SI versions it's hard to track the temporal order with the curved arrows.**

→ Yes, we agree. These figures are rather gruesome to see. Based on your suggestion, we have now updated the percent contribution plots to be stacked as in Figure 1D. Other than that, we deemed it simpler to remove the box plots around the Figure for simplicity. We hope that the updated figure is easier to interpret.

**The mass size mode information in Figure S12 is great. Confirms that there are distinct residential combustion sources.**

→ Thank you! That was the goal we had in mind for this figure.

**The externally derived size distributions for the different factors suggest that COA and the traffic factor have similar size distributions. Does that mean that the large difference in the slopes benchmarking against the AMS in Figure 5 (0.016 vs 0.13) is driven almost entirely by composition?**

→ Yes, despite having similarly sized particles, it is likely that the difference in composition between the two factors, i.e., more fragmentation-prone alkanes in road transport, causes a much larger underestimation of HOA, compared to COA. The measurement discrepancies for these two factors have been discussed in detail in the Supplementary Sections S9 and S10. Figure 5 has been modified

accordingly to request from reviewer 2. These correlations are now moved in the supplementary section **Figure S12.**